# Revisiting properties and concentrations of ice nucleating particles in the sea surface microlayer and bulk seawater in the Canadian Arctic during summer

Victoria E. Irish[1], Sarah J. Hanna[1], Yu Xi[1], Matthew Boyer[2], Elena Polishchuk[1], Mohamed Ahmed[3], Jessie Chen[1], Jonathan P. D. Abbatt[4], Michel Gosselin[5], Rachel Chang[2], Lisa A. Miller[6], Allan K. Bertram[1]

[1] Department of Chemistry, University of British Columbia, 2036 Main Mall, Vancouver, BC V6T 1Z1, Canada
[2] Department of Physics and Atmospheric Science, Dalhousie University, Sir James Dunn Building, 6310 Coburg Road, Halifax, Nova Scotia, B3H 4R2, Canada
[3] Department of Geography, University of Calgary, 2500 University Drive, Calgary, AB T2N 1N4, Canada
[4] Department of Chemistry, University of Toronto, 80 St George Street, Toronto, Ontario, ON M5S 3H6, Canada
[5] Institut des sciences de la mer de Rimouski, Université du Québec à Rimouski, 310 Allée des Ursulines, Rimouski, Québec, QC G5L 3A1, Canada
[6] Institute of Ocean Sciences, Fisheries and Oceans Canada, Sidney, BC V8L 4B2, Canada

*Correspondence to:* Allan Bertram (bertram@chem.ubc.ca)

**Abstract.** Despite growing evidence that the ocean is an important source of ice nucleating particles (INPs) in the atmosphere, our understanding of the properties and concentrations of INPs in ocean surface waters remain limited. We have investigated INPs in sea surface microlayer and bulk seawater samples collected in the Canadian Arctic during the summer of 2016. Consistent with our 2014 studies, we observed that INPs were ubiquitous in the microlayer and bulk seawaters; heat and filtration treatments reduced INP activity, indicating that the INPs were likely heat-labile biological materials between 0.22 and 0.02 μm in diameter; there was a strong negative correlation between salinity and freezing temperatures; and concentrations of INPs could not be explained by chlorophyll *a* concentrations. Unique in the current study, the spatial distributions of INPs were similar in 2014 and 2016, and the concentrations of INPs were strongly correlated with meteoric water (terrestrial runoff plus precipitation). These combined results suggest that meteoric water may be a major source of INPs in the sea surface microlayer and bulk seawater in this region, or meteoric water may be enhancing INPs in this region by providing additional nutrients for the production of marine microorganisms. In addition, based on the measured concentrations of INPs in the microlayer and bulk seawater, we estimate that the concentrations of INPs from the ocean in the Canadian Arctic marine boundary layer range from approximately $10^{-4}$ L$^{-1}$ to $< 10^{-6}$ L$^{-1}$ at -10 °C.

## 1 Introduction

Ice-nucleating particles (INPs) are atmospheric particles that catalyse the formation of ice crystals in clouds at warmer temperatures and lower vapour saturations than needed for homogeneous ice nucleation, thereby influencing cloud

properties and potentially impacting the Earth's radiative properties and hydrological cycle (Boucher et al., 2013; Lohmann, 2002; Lohmann and Feichter, 2005; Tan et al., 2016). Only a small subset of atmospheric particles (about 1 in $10^6$) act as INPs (DeMott et al., 2010, 2016). INPs can catalyse the formation of ice by four different mechanisms: contact freezing, condensation freezing, deposition freezing, and immersion freezing. Immersion freezing, which is the focus of this paper,

occurs when an INP immersed in a supercooled water droplet initiates freezing.

One potential source of INPs to the atmosphere is the ocean. Oceans dominate the Earth's surface coverage, and sea spray generates a large fraction of the aerosol mass in the atmosphere (Lewis and Schwartz, 2004). Several pieces of evidence suggest that the ocean is an important source of INPs to the atmosphere. For example, INPs have been measured in seawater and the microlayer (Fall and Schnell, 1985; Irish et al., 2017; Rosinski et al., 1988; Schnell, 1977; Schnell and Vali,

1975, 1976; Wilson et al., 2015) and in the air above the ocean (Bigg, 1973; Rosinski et al., 1986, 1987, 1988). Marine microorganisms and their by-products can also catalyse ice formation (Burrows et al., 2013; Knopf et al., 2011; Rosinski et al., 1987; Wilson et al., 2015). In addition, modelling studies have illustrated that INP concentrations from the ocean can be important when other sources of INPs, such as mineral dust, are low (Huang et al., 2018b; Vergara-Temprado et al., 2017; Yun and Penner, 2013). Sea spray aerosol is generated at the ocean surface (Blanchard, 1964) and varies considerably in

composition, depending on the production mechanism. The production mechanism determines how much of the sea surface microlayer (herein referred to as the microlayer) compared to bulk seawater will be transferred to the sea spray aerosol (Wang et al., 2017). A recent study has shown that the ice nucleating ability of sub-micrometre particles formed from jet drops is more efficient than those formed from film drops (Wang et al., 2017).

Despite growing evidence that the ocean is an important source of INPs in the atmosphere, our understanding of the

properties and concentrations of INPs in the microlayer and bulk seawater remain limited. For example, information on the spatial and temporal distributions of INPs in the microlayer and bulk seawater has not been investigated in sufficient detail. Nevertheless, this type of information is needed to improve predictions of INP emissions to the atmosphere from the ocean.

Recently, we reported the properties and concentrations of INPs in microlayer and bulk seawater samples collected in the Canadian Arctic during the summer of 2014 (Irish et al., 2017). We found INPs were ubiquitous in the microlayer and

bulk seawater. Heat and filtration treatment of the samples indicated that the INPs were likely heat-labile biological materials with sizes between 0.02 and 0.22 μm in diameter. In addition, we found that the freezing activity of the microlayer and bulk seawater samples was inversely correlated with salinity, implying that the INPs were associated with melting sea-ice or terrestrial runoff. We also observed that the freezing temperatures of the microlayer samples were similar to those of the bulk seawater, in almost all cases.

Building on our previous studies, we returned to the Canadian Arctic during the summer of 2016 to further investigate the properties and concentrations of INPs in Arctic Ocean waters. Locations where samples were collected during both years are indicated in Fig. 1, and the detailed sampling dates and locations in 2016 are given in Table 1. By comparing results from 2016 with those from 2014, we investigate whether the properties, concentrations and spatial profiles of the INPs vary from year-to-year at similar locations. In addition, using stable isotopes of oxygen in the water molecules, we

investigated further the possible importance of melting sea-ice and meteoric water (terrestrial runoff plus precipitation) to the INP concentrations. Measured concentrations of INPs in microlayer and bulk seawater were also used to estimate concentrations of INPs in the Arctic marine boundary layer.

## 2 Experimental

## 2.1 Collection methods

During July and August of 2016 samples were collected from the eastern Canadian Arctic on board the CCGS *Amundsen* as part of the NETCARE project (Fig. 1 and Table 1). Information recorded at each sampling station, are provided in Table S1.

In contrast to 2014, when we collected microlayer samples manually using a glass plate sampler (Irish et al., 2017),

in 2016, microlayer samples were collected using rotating glass plates attached to a sampling catamaran (Shinki et al., 2012). At station 1, the sampling catamaran was deployed from a small boat at least 500 m away from the CCGS *Amundsen*. The sampling catamaran was remotely driven at least 20 m away from the small inflatable, rigid-hull boat before the rotating glass plates were activated remotely. A rotation rate of 10 revolutions per minute was used. From station 2 onwards, the remote control of the rotating glass plates on the sampling catamaran failed. Subsequently, the sampling catamaran was kept

on the upwind side of the small inflatable, rigid-hull boat with its engine turned off, at least 500 m away from the CCGS *Amundsen* to avoid contamination, and the glass plates were rotated manually between 11 to 18 revolutions per minute. The microlayer that adhered to the plates from each rotation was scrapped off with fixed Teflon wiper blades into a manifold and then pumped through Teflon tubing into high-density polyethylene (HDPE) Nalgene bottles (ranging from 250 mL to 2 L in volume). The thickness of the microlayer collected was approximately 80 μm based on the rotation rate (between 11 - 18

revolutions per minute), the average volume collected (3 L) and an average collection time (18 minutes). Bulk seawater samples were collected at the same times and locations through Teflon tubing suspended 0.2 m below the sampling catamaran. The bulk seawater was pumped, using peristaltic pumps, into HDPE Nalgene bottles (ranging from 250 mL to 2 L in volume). After collection, the Nalgene bottles containing the microlayer and bulk seawater samples were kept cool in an insulated container. Upon returning to the ship, the samples were sub-sampled into smaller bottles for subsequent analyses.

The glass plates, aluminium manifold, Teflon tubing and all Nalgene bottles were sterilised first with bleach, then cleaned with isopropanol and finally rinsed with ultrapure water. After cleaning, the sampler was further rinsed by collecting then discarding microlayer and bulk seawater for approximately 2 minutes, before samples were retained.

**2.2 Ice nucleation properties of the samples**

**2.2.1 Droplet freezing technique and INP concentrations**

INP concentrations were determined using the droplet freezing technique (DFT; Koop et al., 1998; Vali, 1971; Whale et al., 2015; Wilson et al., 2015). Sub-samples of the microlayer and bulk seawater were kept in 15 mL polypropylene tubes between 1 to 4 °C for a maximum of 4 hours before INP analysis.

In the freezing experiments three hydrophobic glass slides (Hampton Research, Aliso Viejo, CA, USA) were placed directly on a cold stage (Whale et al., 2015) and between 15 to 30 droplets of the sample, with volumes of 1 μL each, were deposited onto each of the glass slides using a pipette. A total of 45 to 90 droplets were analysed for each sample. A chamber with a webcam attached to the top of it was placed over the slides to isolate them from ambient air, and a flow of ultrapure N$_2$ was passed through the chamber as described by Whale et al. (2015). The droplets were cooled at a constant rate of 10 °C/min from 0 °C to -35 °C and the webcam recorded videos of the droplets during cooling. The freezing temperature of each droplet was determined from the recorded videos and the temperature history of the cold stage (Whale et al., 2015). The temperature of the cold stage was calibrated by measuring the melting temperatures of dodecane (-9.57 °C) and octanol (-14.8 °C) (Whale et al., 2015).

The concentration of INPs per unit volume of liquid, *[INP(T)]$_{vol,liq}$*, was determined from each freezing experiment using the following equation (Vali, 1971):

$$[INP(T)]_{vol,liq} = -\ln\left(\frac{N_u(T)}{N_o}\right)N_o \cdot \frac{1}{V}, \tag{1}$$

where $N_u(T)$ is the number of unfrozen droplets at temperature $T$, $N_o$ is the total number of droplets used in the experiment, and $V$ is the volume of all droplets in a single experiment. Equation 1 represents the concentrations of INPs active at temperature, $T$, and has been justified using Poisson's law (Vali, 1971). The use of Eq. 1 assumes that the concentration of INPs active at temperature $T$ is independent of the cooling rate, which is a reasonable approximation for many atmospherically relevant INPs (Murray et al., 2011; Welti et al., 2012; Wheeler et al., 2015; Wright and Petters, 2013).

**2.2.2. Field and laboratory blanks**

Field blanks for the microlayer samples were prepared by running approximately 1 L of ultrapure water for approximately 1 minute over the glass plates, and through the manifold and tubing used to sample the microlayer. Field blanks for the seawater samples were prepared by running approximately 1 L of ultrapure water for approximately 1 minute through the tubing used to sample bulk seawater. These field blanks were used to evaluate cross contamination between different sampling stations. Laboratory blanks were prepared by passing ultrapure water a 0.22 μm filter.

### 2.2.3 Heating and filtration tests

The freezing temperatures of the microlayer and bulk seawater samples were also measured after they had been passed through syringe filters with three different pore sizes (Whatman 10 µm pore size PTFE membranes, Millex–HV 0.22 µm pore size PTFE membranes, and Anotop 25 0.02 µm pore size inorganic Anopore[TM] membranes) (Irish et al., 2017; Wilson et al., 2015). The samples were left for a maximum of 4 hours before filtration followed by INP analysis.

The freezing temperatures of the samples were also measured after they had been heated to 100 °C (Christner et al., 2008; Irish et al., 2017; Schnell and Vali, 1975; Wilson et al., 2015). In this case, samples were stored at -80 °C for less than 6 months and analysed in the laboratory at the University of British Columbia. Before heating the stored samples, they were completely thawed and homogenised by inverting at least ten times. The freezing temperatures were determined after heating the samples at 100 °C for approximately an hour. Separate experiments show that storage of the samples at -80 °C for a maximum of six months does not affect the INP concentrations (see Fig. S1 in the Supplement).

### 2.2.4 Corrections for freezing temperature depression

The measured freezing temperatures were adjusted for the depression of the freezing point by the presence of salts to generate freezing temperatures applicable to salt-free conditions (salinity = 0 g kg$^{-1}$), which is relevant for mixed phase clouds. In short, water activities of the samples were calculated from measured salinities using an Aerosol Thermodynamic Model (http://www.aim.env.uea.ac.uk/aim/aim.php; Friese and Ebel, 2010; Wexler and Clegg, 2002). Next, the water activity of an ice-salt solution at the median freezing temperature was calculated. The freezing temperature at salinity = 0 g kg$^{-1}$ was then calculated from the difference in these two water activities following the procedure of Koop and Zobrist (2009).

The salinities of the microlayer and bulk seawater samples were measured within 10 minutes of sample collection using a hand-held salinity probe (SympHony; VWR, Radnor, PA, USA). The salinities (measured in practical salinity units (psu)) were corrected using a linear fit to salinometer (Guideline Autosal 8400 B) readings on parallel discrete samples. The correction for freezing point depression by the presence of salts based on the measured salinities ranged from 1.2 to 2.6 °C.

### 2.3 Bacterial and phytoplankton abundance

The abundances of heterotrophic bacteria and phytoplankton < 20 µm (i.e., phycoerythrin-containing cyanobacteria, phycocyanin-containing cyanobacteria and autotrophic eukaryotes) were measured by flow cytometry. Duplicate 4 mL subsamples were fixed with glutaraldehyde (Grade I; 0.12 % final concentration; Sigma-Aldrich G5882) in the dark at room temperature for 15 min, flash-frozen in liquid nitrogen and then stored at -80 °C until analysis. Samples for heterotrophic bacteria enumeration were stained with SYBR Green I (Invitrogen) following Belzile et al. (2008) and counted with a BD

Accuri C6 flow cytometer using the blue laser (488 nm). The green fluorescence of nucleic acid-bound SYBR Green I was measured at 525 nm. Archaea could not be discriminated from bacteria using this protocol; therefore, hereafter, we use the term bacteria to include both archaea and bacteria with high nucleic acid (HNA) content and low nucleic acid (LNA) content. SYBR Green I stains all DNA and RNA, but bacteria and archaea are easily discriminated from other organisms (or

detritus or transparent exopolymeric particles) by their size (side scatter) and fluorescence intensity. In addition, autotrophs stained with SYBR Green I are discriminated from heterotrophic bacteria by their chlorophyll *a* fluorescence.

Samples for < 20 μm phytoplankton abundances were analyzed using a CytoFLEX flow cytometer (Beckman Coulter) fitted with a blue (488 nm) and red laser (638 nm), using CytoExpert v2 software. Using the blue laser, forward scatter, side scatter, orange fluorescence from phycoerythrin (582/42 nm BP) and red fluorescence from chlorophyll (690/50

nm BP) were measured. The red laser was used to measure the red fluorescence of phycocyanin (660/20 nm BP). Polystyrene microspheres of 2 μm diameter (Fluoresbrite YG, Polysciences) were added to each sample as an internal standard (Marie et al., 2005; Tremblay et al., 2009).

## 2.4 Stable oxygen isotopes and water volume fractions

To investigate the possible importance of sea-ice melt and meteoric water (terrestrial runoff plus precipitation) to INP concentrations, we determined $\delta^{18}O$ in the samples. Measurements of $\delta^{18}O$ have been used in the past to distinguish between sea-ice melt and meteoric water in the Arctic Ocean (Alkire et al., 2015; Macdonald et al., 1995; Östlund and Hut, 1984; Tan and Strain, 1980). $\delta^{18}O$, a measure of the ratio of oxygen-18 ($^{18}O$) to oxygen-16 ($^{16}O$) in water molecules, is expressed as per mil (‰) deviations from Vienna Standard Mean Ocean Water (V-SMOW):

$$\delta^{18}O = \left( \frac{\left( \frac{^{18}O}{^{16}O} \right)_{sample}}{\left( \frac{^{18}O}{^{16}O} \right)_{standard}} - 1 \right) \times 1000\,\permil, \qquad (2)$$

where standard corresponds to (V-SMOW). Samples were analysed at the GEOTOP-UQAM stable isotope laboratory at the Université du Québec à Montréal using the $CO_2$ equilibration method (Ijiri et al., 2003), where 200 μL of sample was equilibrated with $CO_2$ for 7 h at 408 °C. The $CO_2$ was then analysed on a Micromass Isoprime[TM] universal triple collector mass spectrometer in dual inlet mode with an AquaPrep[TM] system (Isoprime Ltd., Cheadle, UK). Two internal reference

water samples ($\delta^{18}O$ = -6.71 ‰ and −20.31 ‰) were used to normalise the sample data. Uncertainties in replicate measurements are ± 0.05 ‰ (1σ). $\delta^{18}O$-values were determined for all stations, except stations 1, 10, and 11.

From the measured $\delta^{18}O$ values and measured salinities of the samples, the water volume fractions of sea-ice melt ($f_{SIM}$), water volume fractions of meteoric water ($f_{MW}$), and water volume fractions of seawater $(f_{sw})$ were calculated using the following conservation equations (Yamamoto-Kawai et al., 2005):

$$f_{SIM}S_{SIM} + f_{MW}S_{MW} + f_{SW}S_{SW} = S_{obs} \tag{3}$$

$$f_{SIM}\delta^{18}O_{SIM} + f_{MW}\delta^{18}O_{MW} + f_{SW}\delta^{18}O_{SW} = \delta^{18}O_{obs} \tag{4}$$

$$f_{SIM} + f_{MW} + f_{SW} = 1 \tag{5}$$

where $S$ represent salinity and the subscripts *obs*, *SIM*, *MW*, and *SW* represent observed, sea-ice melt, meteoric water, and
seawater, respectively. For $S_{SIM}$, $S_{MW}$, $\delta^{18}O_{SIM}$, and $\delta^{18}O_{MW}$ we assumed 4 g kg$^{-1}$, 0 g kg$^{-1}$, 0.5 ‰, -20 ‰, respectively, in Eqs.
3-4 (Burgers et al., 2017). The values of $S_{SW}$ and $\delta^{18}O_{SW}$ depend on the reference seawater chosen. In our studies the samples
could have been influenced by either Arctic outflow waters ($S_{SW}$ = 33.1 g kg$^{-1}$ and $\delta^{18}O_{SW}$ = -1.53 ‰) or west Greenland
current waters ($S_{SW}$ = 33.5 g kg$^{-1}$ and $\delta^{18}O_{SW}$ = -1.27 ‰) (Burgers et al. (2017). When calculating $f_{SIM}$, $f_{MW}$, and $f_{sw}$ values we
used $S_{SW}$ = 33.3 ± 0.2 g kg$^{-1}$ and $\delta^{18}O_{SW}$ = -1.40 ± 0.13 ‰, which corresponds to average and limits for Arctic outflow waters
and west Greenland current waters.

## 2.5 Chlorophyll *a*

Chlorophyll *a* concentrations for case 1 waters (waters dominated by phytoplankton) were retrieved from the
GlobColour project website (http://globcolour.info, *ACRI-ST, France*). The GlobColour project provides a high resolution,
long time series of global ocean colour by merging data from several satellite systems. The data used here include retrievals
from either or both the Moderate Imaging Spectrometer (MODIS) on the Aqua Earth Observing System (EOS) mission and
the Visible/Infrared Imager Radiometer Suite (VIIRS) aboard the Suomi National Polar-orbiting Partnership satellite. For
this work we used data merged with weighted averaging, where weightings are based on the sensor and/or product. For more
information regarding the weighted averaging refer to the GlobColour Product User Guide
(http://www.globcolour.info/CDR_Docs/GlobCOLOUR_PUG.pdf). In this study 8-day data were used to achieve the best
balance between spatial coverage (1/24°, ~4 km) and high time resolution. For the chlorophyll *a* concentration at a given
sampling location, we used the grid cell corresponding to the location of that station. We determined the chlorophyll *a*
concentration at all stations except station 8.

Chlorophyll *a* concentrations were also measured in collected samples of seawater. Samples were filtered onto
Whatman GF/F glass-fibre filters, and Chlorophyll *a* concentrations were measured using a Turner Designs AU-10
fluorometer, after 24 h extraction in 90% acetone at 4 °C in the dark (acidification method: Parsons et al. 1984).

# 3 Results and Discussion

## 3.1 Concentrations of INPs

The fraction frozen curves for all microlayer and bulk seawater samples measured in 2016 are shown in Fig. 2. Also shown for comparison are the fraction frozen curves of the samples after filtration through a 0.02 μm Anotop 25 syringe filter, the fraction frozen curves for the laboratory blanks (ultrapure water passed through a filter with a 0.22 μm pore size), and fraction frozen curves for field blanks (ultrapure water passed through the sampling catamaran). The laboratory blanks are at similar or warmer temperatures than the samples passed through a 0.02 μm Anotop 25 syringe filter. Differences are most likely due to the difference in pore sizes of the filters used: the laboratory blanks were passed through filters with a 0.22 μm pore size whereas the samples were passed through filters with a 0.02 μm pore size. Previous experiments in our laboratory have shown that ultrapure water passed through a filter with a 0.02 μm pore size can freeze at slightly colder temperatures than ultrapure water passed through a filter with a 0.22 μm pore size (Fig. S2).

For the bulk seawater, all untreated samples froze at temperatures warmer than the laboratory and field blanks. Freezing temperatures as warm as -6 °C were observed. These results indicate the presence of ice-active material in all bulk seawater samples. For the microlayer samples, all samples froze at temperatures warmer than laboratory blanks. In addition, most samples froze at temperatures warmer than the field blanks. These results also indicate that most microlayer samples contained ice-active material. For some of the samples, the freezing temperatures of the field blanks were warmer than the freezing temperatures of the samples. However, the freezing temperatures of the field blanks should be viewed as an upper limit to the background freezing temperatures, since prior to collecting the field blanks, the sampler had not been rinsed as thoroughly as before collecting the microlayer samples. For the remainder of this paper we will only show and discuss freezing data that were at warmer temperatures than the field blanks.

In Fig. 3 the concentrations of INPs, *[INP(T)]$_{vol,liq}$*, measured in 2016 are compared with concentrations measured in 2014 (sample locations for both years shown in Fig. 1). In both 2016 and 2014, the concentrations of INPs vary by at least 2 orders of magnitude at a given temperature, but warmer freezing temperatures were observed in 2016 compared to 2014.

Figure S3 shows the correlation between $T_{10}$-values (temperatures at which 10 % of the droplets froze) in the microlayer and bulk seawater samples from 2016. We focus on $T_{10}$-values to be consistent with our previous studies and because $T_{10}$-values of the samples were at warmer temperatures than the field blanks in almost all cases. Pearson correlation analysis was applied to compute correlation coefficients (r). P values were also calculated to determine the significance of the correlations at the 95 % confidence level ($p < 0.05$). A strong positive correlation ($r = 0.89$ and $p < 0.001$) was observed between the $T_{10}$-values of the microlayer and the $T_{10}$-values of the bulk seawater, consistent with our previous observations (Irish et al., 2017).

In 2016, 4 out of 9 samples had warmer $T_{10}$-values in the microlayer compared to bulk seawater (Fig. S3). However, in the 2014 samples, only 1 out of 8 samples had warmer $T_{10}$-values in the microlayer compared to bulk seawater (Irish et al., 2017). The difference between 2016 and 2014 may simply be due to year-to-year variations in the properties of

the microlayer relative to the bulk seawater related to variations in oceanic conditions. For example, Collins et al. (2017) documented differences in the activity of marine microbial communities between our 2016 and 2014 campaigns in the Canadian Arctic. In addition, the differences between 2016 and 2014 may be related to sampling techniques. In 2014 the glass plate technique collected a layer that was up to 220 µm thick. In contrast, in 2016 a thinner layer (approximately 80 µm thick) was collected. In the thicker layers collected in 2014, the microlayer INPs would have been diluted by bulk waters by roughly a factor of 2.8. Additional studies of how INP activity varies as a function of microlayer sample thickness are necessary to resolve this issue.

## 3.2 Effect of heating and filtering the samples

Figure S4 shows that the fraction frozen curves were shifted to colder temperatures after the microlayer and bulk seawater samples were heated to 100 °C. These results are similar to what we observed for the 2014 samples (Irish et al., 2017). This suggests that the ice-active material we found in the microlayer and bulk seawater samples was likely heat-labile biological material (Christner et al., 2008).

Figure S5 shows that the temperature at which droplets froze in microlayer and bulk seawater samples significantly decreased after the samples were passed through a 0.02 µm filter, but not through 10 µm or 0.22 µm filters. A similar result was observed in the 2014 samples (Irish et al., 2017), suggesting that the INPs in the microlayer and bulk seawater were between 0.22 µm and 0.02 µm in size.

## 3.3 Spatial distributions of INPs in the Canadian Arctic

The spatial distributions of $T_{10}$-values for bulk seawater samples in both 2016 and 2014 are shown in Fig. 4. The spatial distributions are similar for microlayer samples (Fig. S6). In each panel the colour scales have been adjusted to easily compare the general pattern of $T_{10}$-values between years. For both 2014 and 2016, the $T_{10}$-values for samples taken from northern Baffin Bay and Nares Strait between Greenland and Canada, above 75 °N, are generally lower than the $T_{10}$-values elsewhere. To further investigate the similarities in spatial patterns between 2014 and 2016, we compared $T_{10}$-values at sampling sites in close proximity for the two years (Fig. 5a). A strong positive correlation ($r = 0.93$, $p < 0.001$) was found between the $T_{10}$-values measured in 2014 and $T_{10}$-values measured in 2016 at those proximal locations (Fig. 5b), suggesting that the general spatial distribution of $T_{10}$-values measured in 2014 and 2016 were similar even though warmer freezing temperatures were observed in 2016 compared to 2014 (Fig. 3).

## 3.4 Correlations with biological, chemical, and physical properties of the bulk seawater

In Table 2 and Fig. S7, we present correlations between $T_{10}$-values for bulk seawater in 2016 and heterotrophic bacterial abundance, phytoplankton (including 0.2-20 µm photosynthetic eukaryotes and cyanobacteria) abundance, salinity, and temperature. The strongest correlation was with salinity (r = -0.83, p ≤ 0.001). Similar correlations were observed for $T_{50}$-values (Table S2). One possible explanation for the negative correlation between $T_{10}$-values and salinity is a non-colligative effect of sea salt on the freezing temperature. For example, solutes can impact freezing temperature by blocking INP active sites (Kumar et al., 2018). To test this hypothesis, we varied the salinity in one of the microlayer samples (Station 4) by adding commercial sea salt (Instant Ocean™), while keeping the concentration of ice nucleating material in the samples constant (see Supplement Section S1for more details). As the salinity of the sample was increased from 29 to 55 g kg$^{-1}$, the $T_{10}$-values for the salinity-enhanced samples (after correcting for freezing point depression) varied by less than the uncertainty in the measurements (Fig. S8 in the Supplement). These results suggest that sea salt does not have a non-colligative effect on the freezing temperature of the samples, at least not for the microlayer sample tested (Station 4). Consistent with these results, non-colligative effects have not been observed in previous studies of immersion freezing with seawater and sodium chloride solutions (Alpert et al., 2011a, 2011b; Knopf et al., 2011; Wilson et al., 2015; Zobrist et al., 2008). Non-colligative effects have been observed in immersion freezing studies with ammonium containing salts, but these results are not likely relevant for seawater solutions (Whale et al., 2018).

As suggested in our earlier study (Irish et al., 2017), another possible explanation for the negative correlation between salinity and freezing temperature is that the INPs are associated with either sea-ice melt or terrestrial runoff (including that from melting glaciers or permafrost). Melting sea-ice and terrestrial runoff have lower salinities than seawater. In addition, sea-ice melt and terrestrial runoff often contain microorganisms and their exudates, which can be especially effective INPs (Assmy et al., 2013; Boetius et al., 2015; Christner et al., 2008; Ewert and Deming, 2013; Fernández-Méndez et al., 2014). Terrestrial runoff could also enhance the production of INPs in the ocean by providing additional nutrients for the growth of marine microorganisms.

Figure 6 shows the $T_{10}$-values of bulk seawater as a function of the water volume fraction of meteoric water ($f_{MW}$) and water volume fraction of sea-ice melt ($f_{SIM}$) calculated using Eqs. 3-5. A strong positive correlation (r = 0.91, p < 0.001) was observed between $T_{10}$ and $f_{MW}$ in the samples. In contrast, the correlation between $T_{10}$ and $f_{SIM}$ in the samples was weaker and the p-value was close to 0.05 (r = 0.63, p = 0.048). These combined results suggest that meteoric water (terrestrial runoff plus precipitation) may be a major source of INPs in this area, or alternatively meteoric water may be enhancing INPs in this area by providing additional nutrients for the production of marine microorganisms. Terrestrial runoff has also been identified as a major source of INPs in temperate rivers and lakes (Knackstedt et al., 2018; Larsen et al., 2017; Moffett et al., 2018).

### 3.4.1 Chlorophyll *a* correlations

Figure 7 shows correlations between the chlorophyll data retrieved from GlobColour and the $T_{10}$-values for the microlayer and bulk seawater. The correlations between $T_{10}$-values in the microlayer or bulk seawater and chlorophyll *a* are not statistically significant. Figure S9 shows the relationship between the measured chlorophyll *a* concentrations and the $T_{10}$-values for the microlayer and bulk seawater. Again, the correlations are not statistically significant. Our results from satellite and our measured chlorophyll *a* data are consistent with recent work by Wang et al. (2015), who showed that INP concentrations in sea spray aerosol emitted during a mesocosm tank experiment were not simply coupled to chlorophyll *a* concentrations.

### 3.5 Predictions of INP concentrations in the Arctic marine boundary layer

In the following, we provide an initial estimate of the concentration of INPs in the Arctic marine boundary layer based on our freezing results. First, we calculated the concentration of INPs in the liquid per unit mass of sea salt, *[INP(T)]*$_{mass,salt}$, using the following equation:

$$[INP(T)]_{mass,salt} = -\ln\left(\frac{N_u(T)}{N_o}\right) N_o . \frac{1}{V\rho S} \tag{6}$$

where $\rho$ is the density of water and $S$ is the salinity of the seawater. Plots of *[INP(T)]*$_{mass,salt}$ as a function of temperature calculated from our freezing results are shown in Fig. S10. From *[INP(T)]*$_{mass,salt}$, the concentration of INPs per unit volume of air in the Arctic marine boundary layer, *[INP(T)]*$_{vol,air}$ was estimated with following equation:

$$[INP(T)]_{vol,air} = [INP(T)]_{mass,salt} . c \tag{7}$$

where $c$ is the concentration of sea salt in the Arctic marine boundary layer. Average concentrations of sea salt at Barrow, Alaska (71.3° N, 156.6° W), Alert, Nunavut, Canada (82.5° N, 62.5° W), and Zeppelin, Svalbard, Norway (78.9° N, 11.9° E) are 1.5, 0.1, and 0.6 µg m$^{-3}$ in July, and 1.4, 0.1, and 0.5 µg m$^{-3}$ in August, respectively (Huang et al., 2018a). For these exploratory calculations we used a value of 1 µg m$^{-3}$, which is within the range of the concentrations mentioned above.

Shown in Fig. 8 are estimated values for *[INP(T)]*$_{vol,air}$ based on our freezing data and a concentration of sea salt in the Arctic marine boundary layer of 1 µg m$^{-3}$. Based on our freezing data, *[INP(T)]*$_{vol,air}$ ranges from ~$10^{-4}$ L$^{-1}$ to < $10^{-6}$ L$^{-1}$ for freezing temperatures ranging from -5 °C to -10 °C. Over this temperature range, the highest estimated values for *[INP(T)]*$_{vol,air}$ were associated with two microlayer samples, and only these two microlayer samples resulted in *[INP(T)]*$_{vol,air}$ values as high as observed in direct atmospheric measurements of *[INP(T)]*$_{vol,air}$ in the marine boundary layer (Fig. 8) (DeMott et al., 2016; Irish et al., 2019). For freezing temperatures ranging from -15 °C to -25 °C, our estimated *[INP(T)]*$_{vol,air}$ values range from >$10^{-4}$ L$^{-1}$ to < $10^{-6}$ L$^{-1}$. Over this temperature range, many of our samples result in *[INP(T)]*$_{vol,air}$ values much less than observed in direct atmospheric measurements (Fig. 8). However, since our estimated *[INP(T)]*$_{vol,air}$ values are limited to $2 \times 10^{-6}$ L$^{-1}$, we cannot determine if our most active samples give *[INP(T)]*$_{vol,air}$ values

similar to direct atmospheric measurements for freezing temperatures of -15 °C to -25 °C. The following caveats should be kept in mind when interpreting Fig. 8: first, we did not consider the possible enrichment of INPs in sea salt aerosols compared to the microlayer or bulk seawater samples, which can result from the bubble bursting mechanism. Second, the concentrations of sea salt used to estimate $[INP(T)]_{vol,air}$ was likely an upper limit based on the previous measurements at

Barrow, Alert and Zeppelin.

## 4 Summary and conclusions

The INP concentrations in microlayer and bulk seawater samples were determined at eleven stations in the Canadian Arctic during the summer of 2016 and compared to measurements made in 2014 (Irish et al., 2017). Filtration reduced the freezing temperatures of all samples, suggesting ice-active particulate material was universally present in the

microlayer and bulk seawaters we studied. Some samples had freezing temperatures as high as -5 °C. Freezing temperatures also decreased after heat treatment, indicating that the ice-active material was likely heat-labile biological material, consistent with previous measurements of INPs in the microlayer (Wilson et al., 2015) and bulk seawater (Schnell, 1977; Schnell and Vali, 1975, 1976). The ice-active material we observed was between 0.22 μm and 0.02 μm in size, also consistent with previous studies of INPs in the microlayer (Wilson et al., 2015) and bulk seawater (Rosinski et al., 1986;

Schnell and Vali, 1975).

We found similar spatial distribution of INPs in both years. In 2016, however, we observed generally higher concentrations of INPs nucleating ice at higher temperatures, particularly in the microlayer samples. This could, in part, be because we sampled a thinner microlayer in 2016, a hypothesis that could be tested by collecting microlayer samples using both collection methods in the same region at the same time. The observed differences could also simply be a result of

variability in oceanographic conditions between the two expeditions.

We observed a strong positive correlation between $T_{10}$-values and the volume fraction of meteoric water in the bulk seawater samples. These results suggest that meteoric water may be a major source of INPs in Arctic coastal regions. Alternatively, meteoric water may be enhancing INPs in this area by providing additional nutrients for the production of marine microorganisms. Related, recent studies have measured high concentrations of INPs in freshwater sources such as

rivers and lakes in other parts of the world (Knackstedt et al., 2018; Larsen et al., 2017; Moffett et al., 2018).

Exploratory calculations, using our freezing data, suggest that the concentrations of INPs from the ocean in the marine boundary layer range from $\sim 10^{-4}$ $L^{-1}$ to $< 10^{-6}$ $L^{-1}$ at -10 ° C. Furthermore, only the most active samples we studied gave calculated INP concentrations as high as observed in previous measurements of INPs in the marine boundary layer (DeMott et al., 2016; Irish et al., 2019). However, these exploratory calculations have caveats that need to be considered in

future studies.

**Data availability**

Underlying material and related items for this manuscript are located in the Supplement.

**Author contribution**

AKB, JPDA, LAM, and VEI conceptualised the research. VEI, MB, MA, and RC collected the samples. SJH, YX, MG, LAM, and MA provided additional data. VEI analysed the data. VEI, SJH, MG, LAM, and AKB wrote the publication. All co-authors reviewed the paper.

**Competing interests**

The authors declare that they have no conflict of interest.

**Acknowledgements**

We thank the scientists, officers, and crew of the CCGS *Amundsen* for their support during the 2014 and 2016 expeditions; Lucius Perreault for land-based support with the microlayer sampler; Allison Lapin, Eugene Shen, and Hang Nguyen for help with INP analyses; Joannie Charette, Aude Boivin-Rioux, and Claude Belzile for flow cytometry analyses; and Tonya Burgers for help with $\delta^{18}O$ data collection, analysis, and interpretation. We would also like to thank the Natural Sciences and Engineering Research Council of Canada (the NETCARE project), Fisheries and Oceans Canada and ArcticNet (Network of Centres of Excellence of Canada) for funding this work. GlobColour data (*http://globcolour.info*) used in this study were developed, validated, and distributed by ACRI-ST, France.

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

| Station number | Sampling start time (UTC) | Location |
| --- | --- | --- |
| Station 1 | 20th July 2016 18:30 | 60°17.921N 062°10.750W |
| Station 2 | 28th July 2016 15:30 | 67°23.466N 063°22.067W |
| Station 3 | 1st August 2016 13:30 | 71°17.200N 070°30.236W |
| Station 4 | 6th August 2016 13:30 | 76°20.341N 071°11.418W |
| Station 5 | 8th August 2016 11:00 | 76°43.777N 071°47.267W |
| Station 6 | 9th August 2016 14:30 | 76°18.789N 075°42.963W |
| Station 7 | 11th August 2016 17:00 | 77°47.213N 076°29.841W |
| Station 8 | 13th August 2016 10:30 | 81°20.041N 062°40.774W |
| Station 9 | 15th August 2016 14:00 | 78°18.659N 074°33.757W |
| Station 10 | 21st August 2016 10:00 | 68°19.199N 100°49.010W |
| Station 11 | 23rd August 2016 10:30 | 68°58.699N 105°30.022W |

**Table 1.** Sampling times and geographic coordinates for the eleven stations investigated.

|  | Bulk $T_{10}$-value | | |
|---|---|---|---|
|  | r | p | n |
| Heterotrophic bacterial abundance | **-0.77** | **0.003** | **11** |
| Total phytoplankton abundance (0.2 - 20 μm) | 0.19 | 0.287 | 11 |
| Salinity | **-0.83** | **0.001** | **11** |
| Temperature | 0.20 | 0.285 | 10 |

**Table 2.** Correlations between biological and physical properties of bulk seawater and $T_{10}$-values for 2016. Values in bold indicate results that are statistically significant.

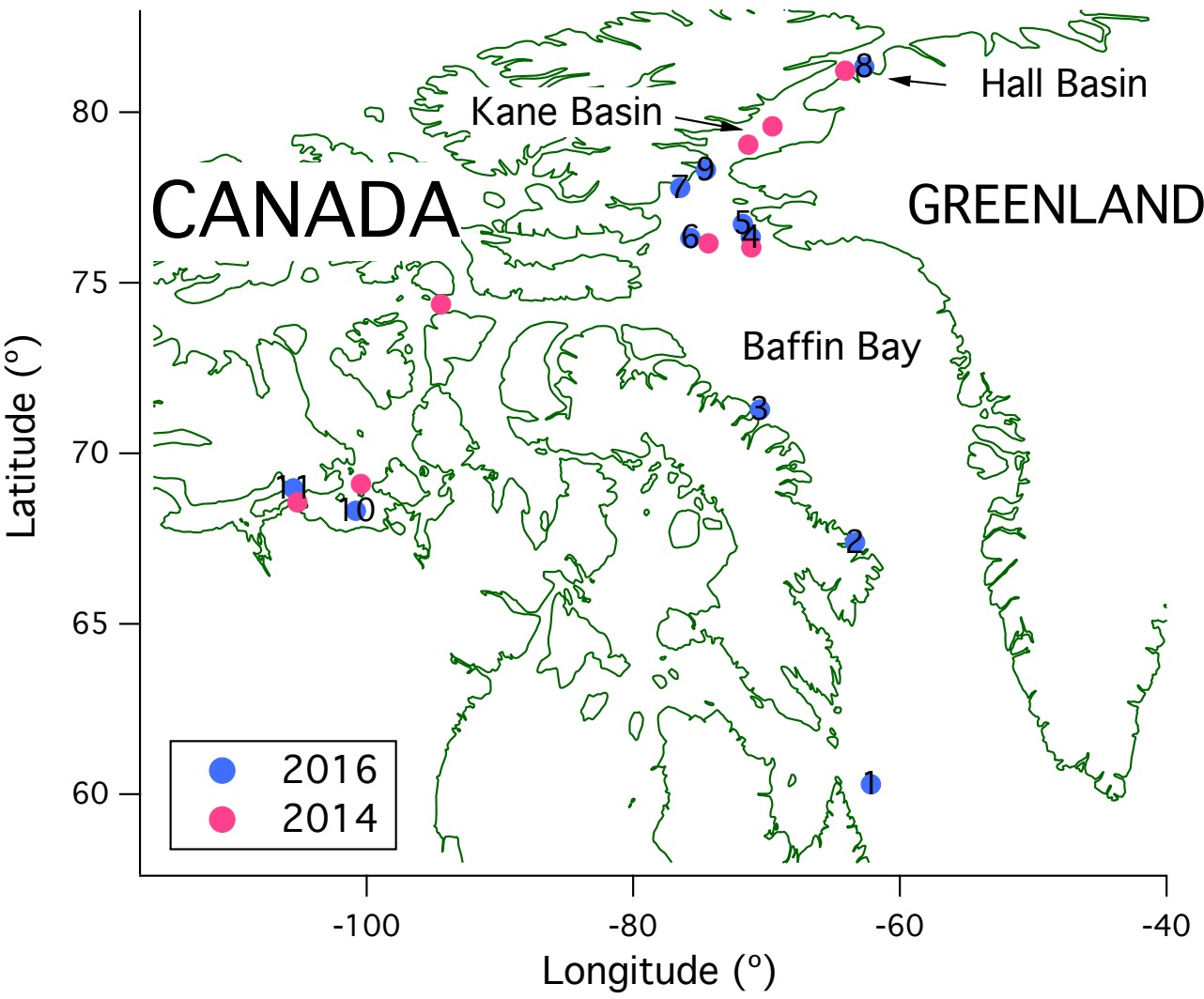

**Figure 1.** Map showing locations of microlayer and bulk seawater sampling in 2014 (pink) and 2016 (light blue with specific station numbers in black).

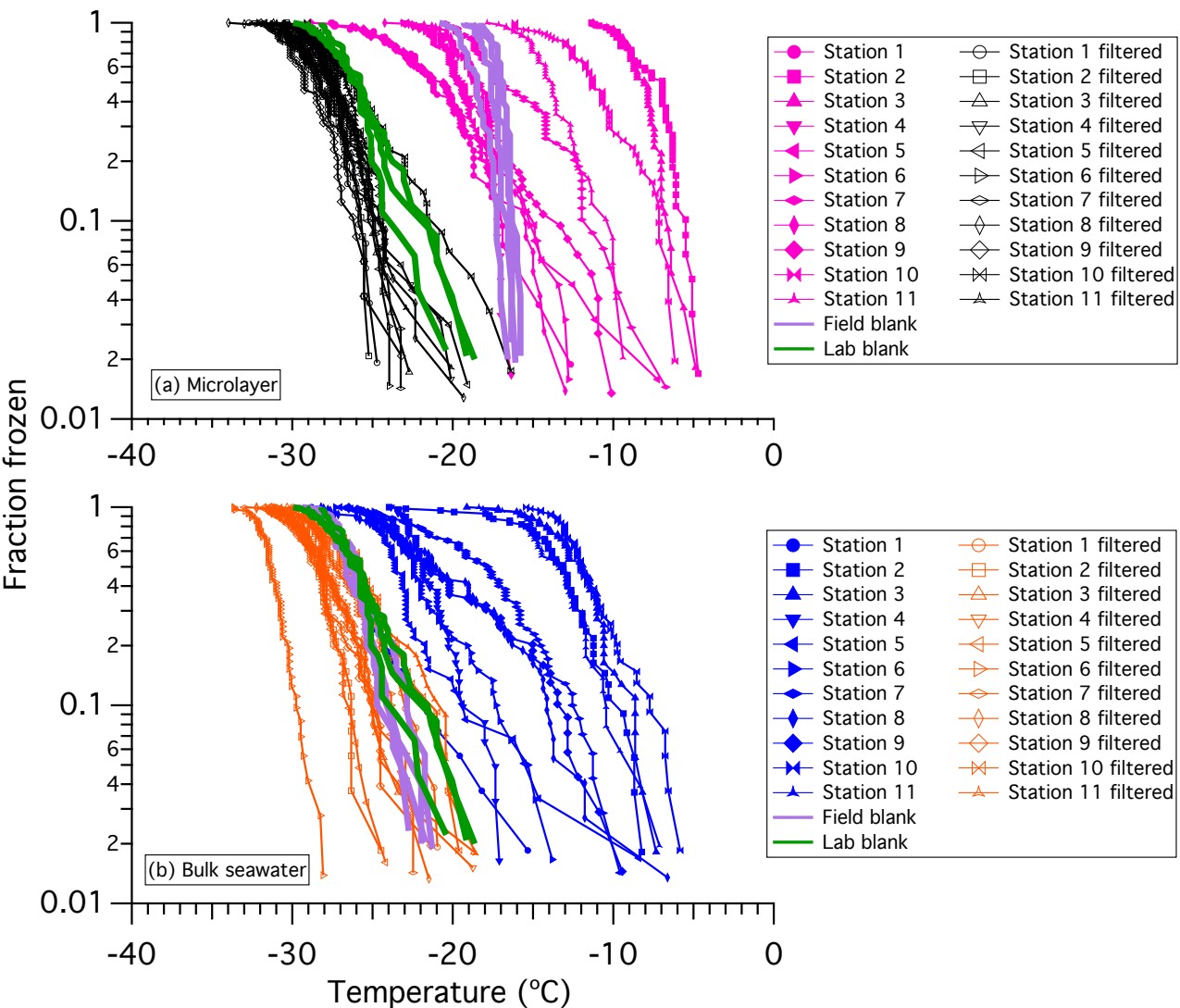

**Figure 2.** Fraction of droplets frozen (in the immersion mode) versus temperature for (a) the microlayer, and (b) bulk seawater. Each line shows the results for 3 replicate experiments of a sample or a sample passed through a 0.02 µm filter, with a total of between 45 to 60 freezing events in each set. Each data point corresponds to a single freezing event in the experiments. Also included are the laboratory blanks (ultrapure water passed through a 0.22 µm filter), and the field blanks (ultrapure water sampled through the sampling catamaran). All microlayer and bulk seawater freezing points were corrected for freezing point depression to account for dissolved salts in seawater (Section 2.2.3). The uncertainty in temperature is ± 0.3 °C.

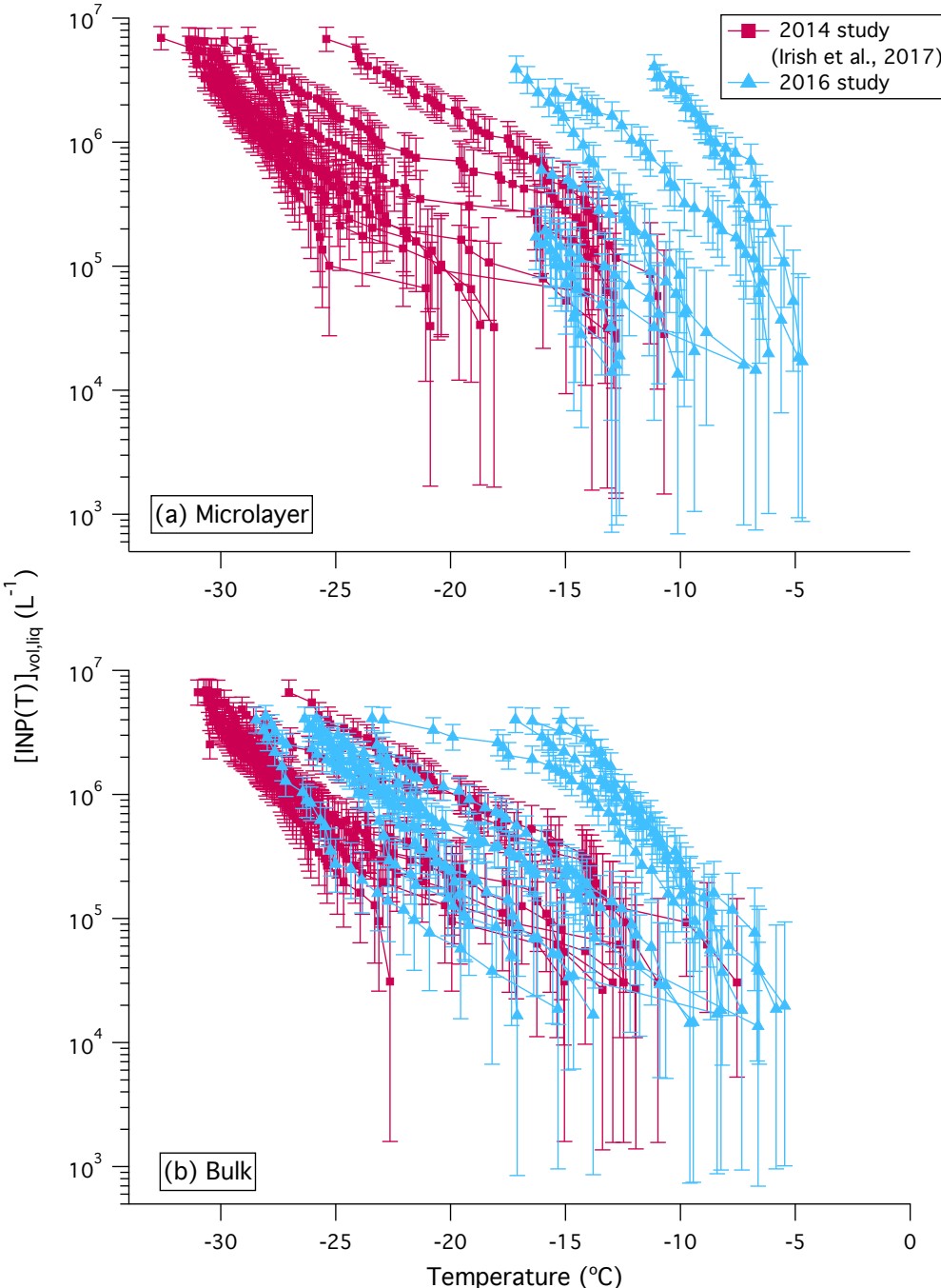

**Figure 3.** Comparison of the concentrations of INPs, *[INP(T)]*$_{vol,liq}$, in (a) the microlayer and (b) bulk seawater samples from the 2014 (pink squares) and 2016 (blue triangles) studies. All data were corrected for freezing point depression. Error bars represent the statistical uncertainty due to the limited number of nucleation events observed in the freezing experiments (Koop et al., 1997). Only freezing data that was at warmer temperatures than the field blanks are included.

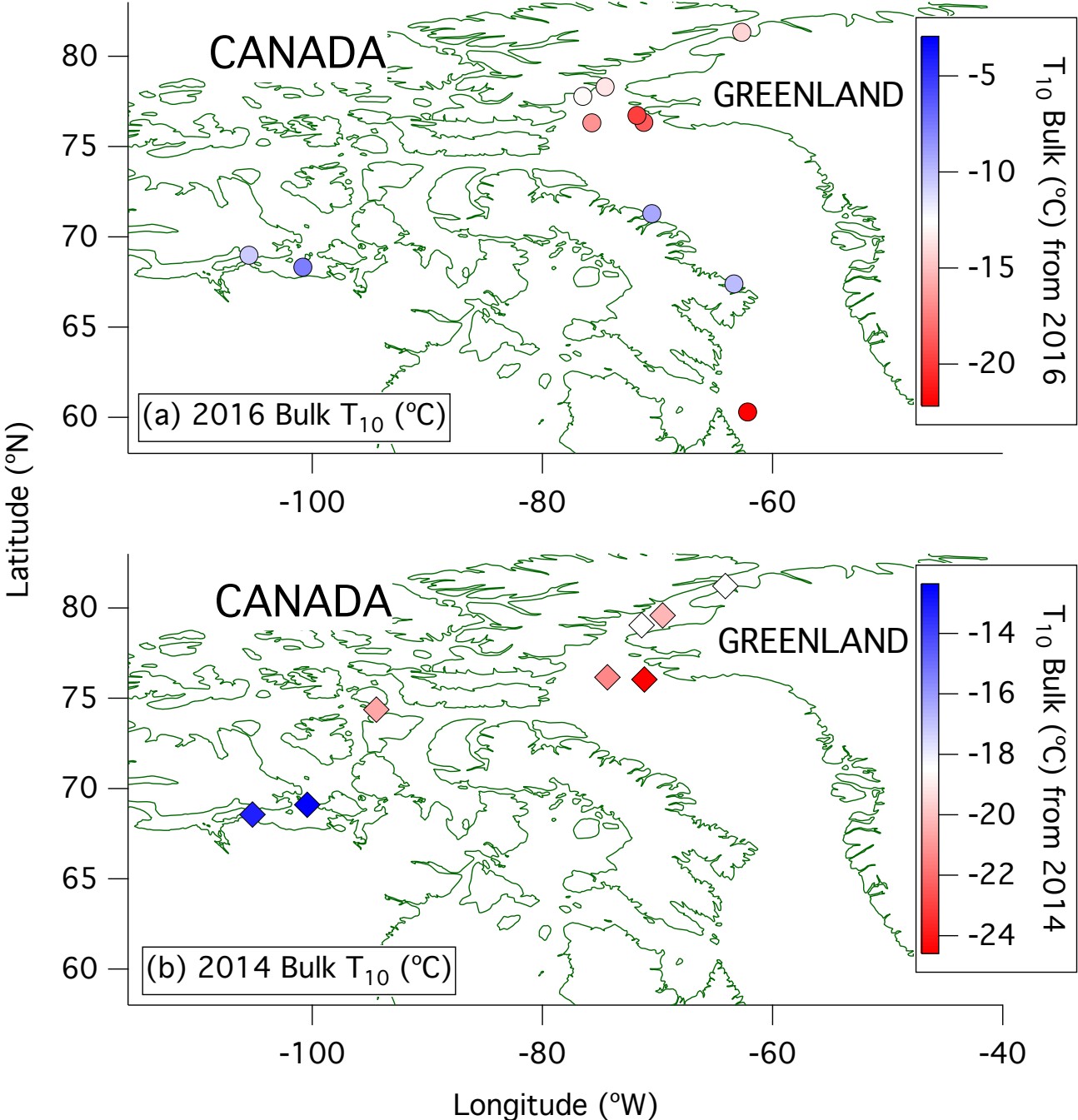

**Figure 4.** Spatial distributions of $T_{10}$-values in (a) 2016 and (b) 2014 for bulk seawater.

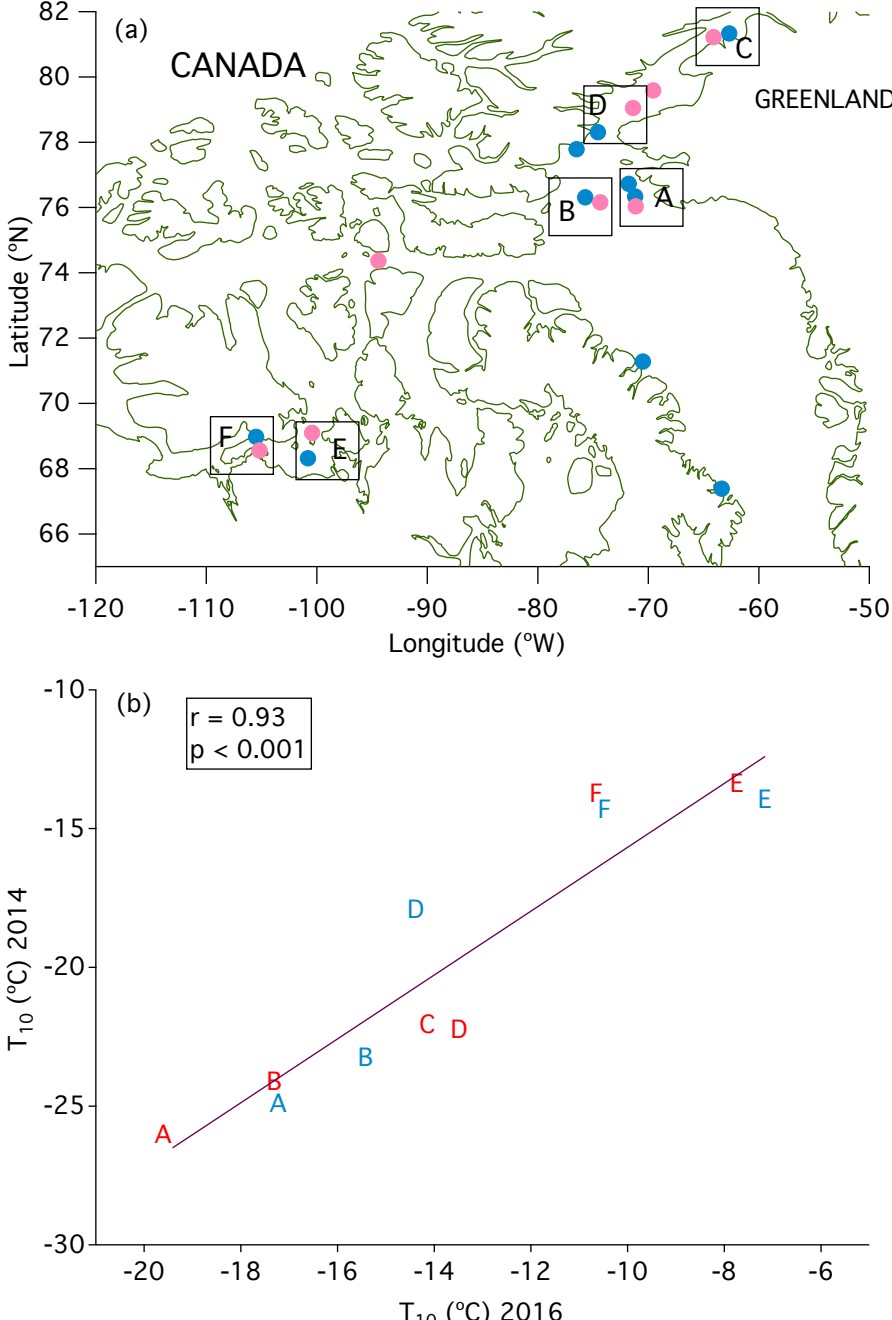

**Figure 5.** (a) Map showing regions of similar sampling locations in 2014 (pink) and in 2016 (blue). Sampling sites in 2014 that were near sampling sites in 2016 were paired together (indicated with boxes in the figure) and assigned letters A-F. Although there are two stations in box A for 2016, we only used data for the station that was closest to the one in 2014. (b) Relationships between $T_{10}$-values for microlayer and bulk seawater samples in 2014 and 2016 for similar sampling locations. The letters plotted in (b) indicate the locations in (a). Red letters represent bulk seawater data and blue letters represent microlayer data. Only freezing data that was at warmer temperatures than the field blanks are included.

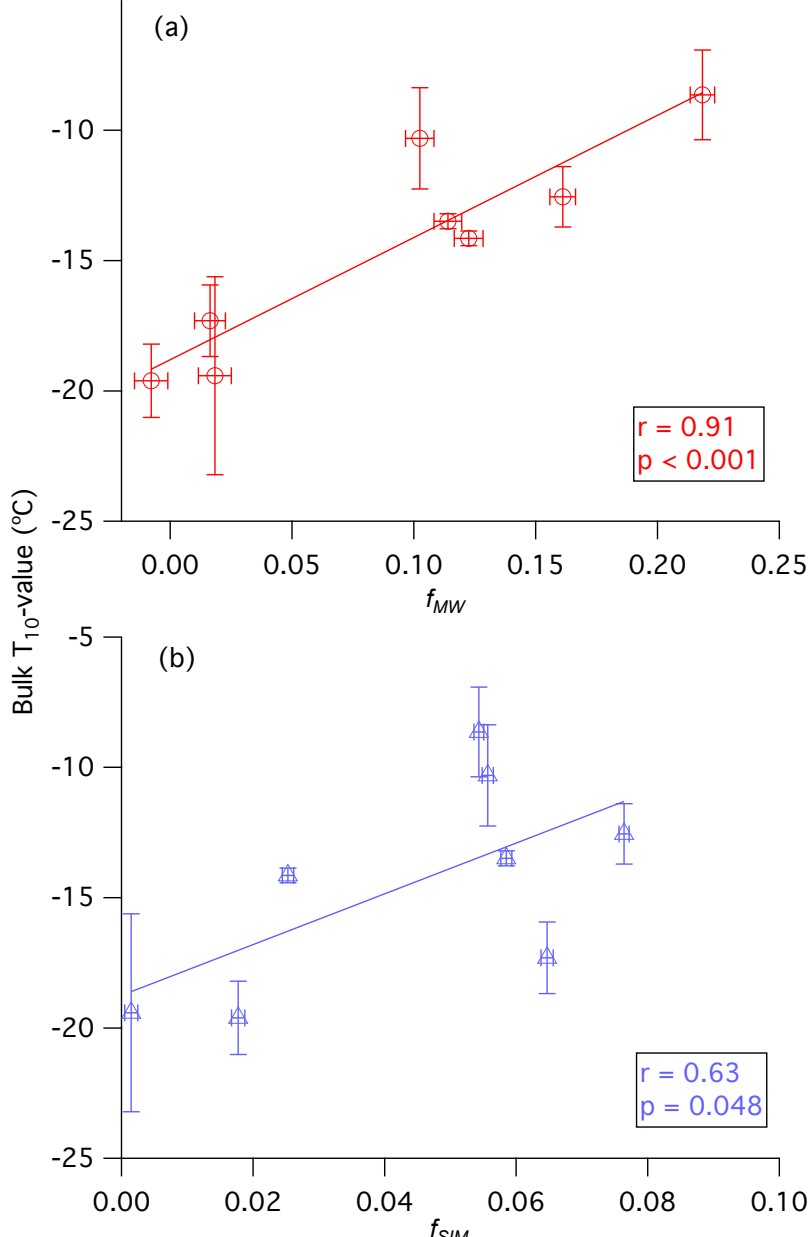

**Figure 6.** Relationships between $T_{10}$-values for bulk seawater and (a) the water volume fractions for meteoric water, $f_{MW}$, and (b) the water volume fractions for sea-ice melt, $f_{SIM.}$. The x-error bars are due to the uncertainties in seawater salinities and seawater $\delta^{18}O$ values used for calculating $f_{MW}$ and $f_{SIM.}$. For further details see Section 2.4. The y-error bars correspond to the 95% confidence interval for three repeat experiments.

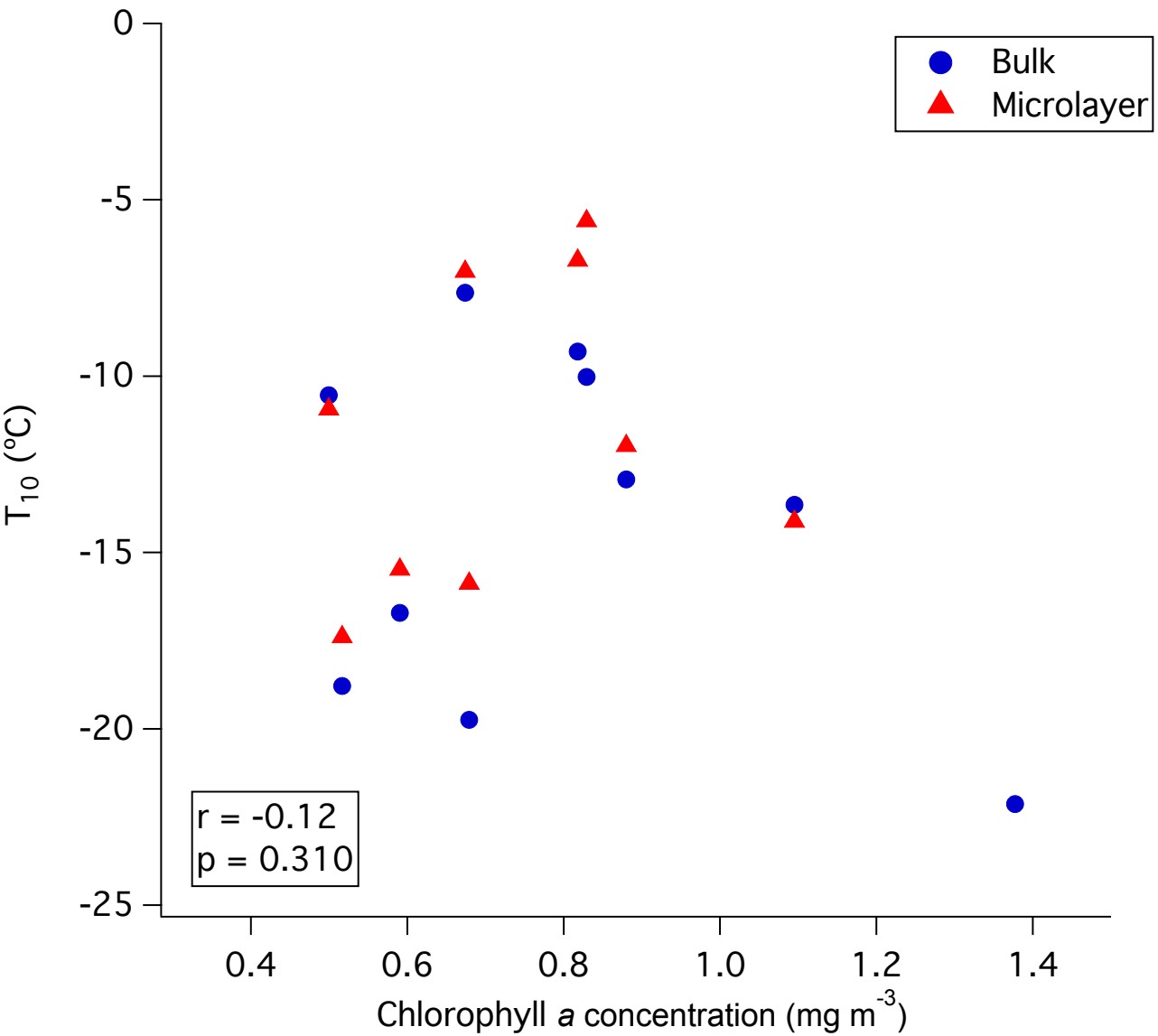

**Figure 7.** Relationship between satellite-derived chlorophyll *a* concentrations, and the $T_{10}$-values of microlayer and bulk seawater for 2016. Only freezing data that was at warmer temperatures than the field blanks are included.

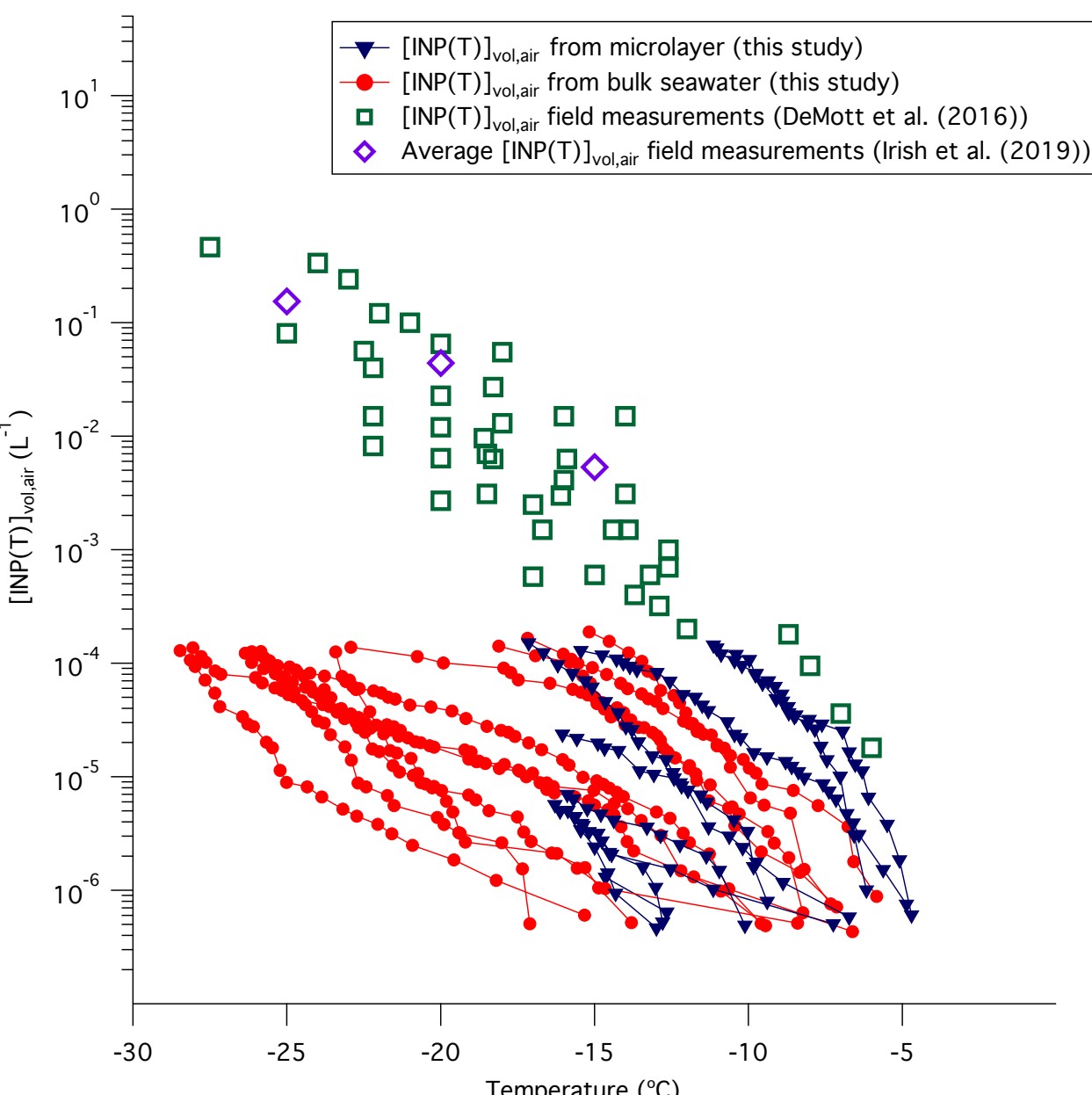

**Figure 8.** Plot of calculated *[INP(T)]ᵥₒₗ,ₐᵢᵣ* as a function of temperature based on our freezing data and an assumed sea salt aerosol concentration of 1 μg m⁻³. Also included are measured *[INP(T)]ᵥₒₗ,ₐᵢᵣ* from several recent field campaigns in the marine boundary layer reported in DeMott et al. (2016) and Irish et al. (2019). Only freezing data that was at warmer temperatures than the field blanks are included.