# Peer review of "Revisiting properties and concentrations of ice nucleating particles in the sea surface microlayer and bulk seawater in the Canadian Arctic during summer"

_Atmospheric Chemistry and Physics, 2018_

## Short Comment (SC1) · 18 Sep 2018

I really enjoyed reading this paper. It is not surprising that there were differences in the IN between years. There could even be differences within a single year. It is important to note that the production mechanism determines how much of the SSML will be transferred to SSA...and there are difs in IN activity of sea spray produced by jet vs film drops (Wang, et al., PNAS, 2017, The role of jet and film drops in controlling the mixing state of submicron sea spray aerosols). The IN efficiency of jet drops has been shown in this publication to be higher than film drops–which goes against the common

[Figure]

assumption that there will be more bioparticles in the SSML than in bulk seawater. Further, the # of jet vs film drops can change over the course of a bloom (see Fig 1 below taken from supp info in above referenced PNAS paper). It is important to remember that physical, biological, and chemical factors all control the formation and composition of SSA and will thus affect the IN activities.

[Figure]

[Figure]

Figure S9. Time series of number concentrations of film drop particles and jet drop particles measured during the phytoplankton bloom. See main text for the method of calculation.

**Fig. 1.** jet vs film drops over course of a phytoplankton bloom

---

## Referee Comment (RC1) · Anonymous Referee #1 · 1 Oct 2018

The manuscript titled "Revisiting properties and concentrations of ice nucleating particles in the sea surface microlayer and bulk seawater in the Canadian Arctic during summer" by Irish et al. presents an ice nucleation study using droplets generated from bulk and surface Arctic seawater. The authors use filtration to estimate the size of the ice nucleating agent. After heating the water, freezing temperatures of droplets decreased. Finally, the authors also measure salinity, numbers of bacteria and phytoplankton cells and correlate them to the temperature at which 10% of the droplets froze, $T_{10}$, in each sample. Chlorophyll satellite data was also correlated with $T_{10}$. Warmer freezing temperatures correlated with decreasing salinity and decreasing bacteria concentrations. The authors also found warmer freezing temperatures in this study compared with the one conducted in 2014.

Overall, I find the manuscript unsuitable for ACP in the present form as I fail to see new findings in chemistry and physics. Furthermore, the results and data analysis are essentially identical to Irish *et al.* (2017) with the only difference of the sampling depth due to use of a rotating drum. Granted, the data here is new because it is taken at a later year and with a well defined scientific approach. However, the results and analysis in this manuscript are essentially copies of the author's previous paper without any scientific extension. In my major comments below, I have introduced some ideas to extend their work. As it is now, I do not find any conclusions anywhere in the manuscript and the word is not even written except for the title of the section. Please understand the difference between conclusions and observational results. Overall, I will not recommend publication in ACP.

Major Comments

The authors lack ice nucleation physics. There is no nucleation theory or any application of active sites for comparison with other studies. This is because there is no surface area estimate of insoluble material in their droplets. If water and filters are still available, then total particulate mass or surface area of insoluble particles could be obtained. Forexample, filters can be washed, dried and weighed and water can be used to get a size distribution from the flow cytometry. Another point is that correction for freezing point depression follows a water activity approach that Koop and Zobrist (2009) used for other biogenic ice nucleators. A plot of INP vs. $\Delta a_{\text{w,het}}$ could be made which allows the authors to discusses the effect (or lack thereof) of ionic activity on ice nucleation. $\Delta a_{\text{w,het}}$ could be compared with other biogenic ice nucleators.

The authors lack cloud and atmospheric physics. The authors could use SSA production formulations measured from previous studies to calculate the number of ice

forming particles per liter of air. Vertical and horizontal motion (updraft and 10 m high wind speed) provided from meteorological data, or reanalysis can then be used to give some notion of the total ice nucleating particles in air. Does the $T_{10}$ data or some other percentage of droplets frozen, correspond to a mixed phase cloud base or ice water path from satellite data?

The authors lack ocean physics. There is countless studies documenting the enrichment or lack of enrichment of material in a microlayer with respect to bulk water. These materials can be surfactants, insoluble particles, or other materials such as proteins and polysaccharides. The interesting result from both the present manuscript and Irish *et al.* (2017) is that the ice nucleation ability is the same for bulk and microlayer water. This could mean that the ice nucleating particles are not surface active? What compounds in the ocean are uniformly distributed through the microlayer and bulk water? Are there soluble surfactants in bulk water that are transported to the microlayer? Is there a difference in surface tension between microlayer and bulk water?

p.6 l.26 - p.7 l.1-5: Clearly, freezing temperatures warmer than pure water indicate heterogeneous droplet freezing. However, the "procedural blank" resulted in freezing temperatures at -16° C. How is it possible that freezing was observed below this temperature? On l.4-5 the answer is given that rinsing times were different (a fact not mentioned in the experimental section), so the freezing temperatures of the "blank were due to (cross-)contamination. How can we then compare any measurements of these to the blanks? In microlayer samples, Fig. S2a shows that no data below -16° C can be trusted. If these were the blanks for the experiments, the freezing curves should follow exactly the procedural blank data which would be seen as a discontinuity (step in the graph) of the freezing temperature around -16° C. This is not the case and so I would conclude that this blank has nothing to do with the data at all and suggest there is no blank experiment for these data using the same procedures. How is this data at all trustable? I now understand why the authors use $T_{10}$ and not median freezing as reported in Koop and Zobrist (2009), because if they did there would be

no difference with their freezing points of microlayer water and the blanks. I am very concerned that freezing temperatures were due to cross contamination because of the lack of reproducibility for the blanks, as the freezing temperatures of the microlayer and bulk seawater do not follow the blanks at all.

p.6 l.26 - p.7 l.1-5: In the same section I find that freezing temperatures of filtered water (through the sampler) are less than ultra pure water (not through the sampler) by about 5-10° C. I doubt the seawater was more pure than the ultrapure water, so what is wrong here? Are the authors certain of the freezing point correction with the E-AIM model? Is there an uncertainty of $\pm 5 - 10°$ C? I cannot accept this result and it makes me seriously doubt the accuracy of these experiments. The blank should be the lowest freezing temperature.

Figure S3: The ultrapure water data here is about 5° C different from the ultrapure water in Fig. S2. This indicates to me that the authors experiment is reproducible to $\pm 5°$ C. This is a large uncertainty which is not stated in the paper.

p.3 l.12: It is not possible to name your instrument as an autosampler when for the majority of the stations the authors had to manually rotate the drum.

Equation 1: How does this equation account for the possibility of multiple INP's? Does the author observe more than one nucleation event in a droplet before it crystalizes? How can they tell if the droplet has 1 or 100 INPs inside? This method of analysis is 45 years old, do the authors have an updated analysis for quantifying freezing?

p.8 l.23-31: The logic is flawed here. Melting sea ice decreases salinity and releases bacteria to the ocean (p.24-26). Decreasing salinity yields warmer $T_{10}$ (Fig. 6 lower left). Decreasing salinity yields increasing bacteria (p.8 l.24-27). Finally, increasing bacteria yield lower $T_{10}$ (Fig. 6 upper left). So why do I read in p.8 l.30-31 that bacteria are fewer in melting sea ice and that bacteria increasing ice nucleation ability? This argument is highly contradictory.

The most major problem I see in the manuscript is that it is a copy of the authors previous manuscript. The majority of section 4 is a repeat of Irish *et al.* (2017). The last paragraph of section 4 states that the only new finding is that concentrations are higher in 2016 than in 2014, but dismisses this finding due to a different sampler. This study ended in August 2016, but the Irish *et al.* (2017) paper (using only 2014 data) was submitted April 2017. Why wasn't the data presented in this manuscript used in Irish *et al.* (2017)? In any case, the authors should extend their work with new data and new discussion that includes physical and chemical understanding before I recommend publication in ACP.

Minor Comments

What are the "properties" of ice nucleating particles? How is that different from "freezing properties"? How is that different from "ice nucleation properties"? Properties of the microlayer? This word is used countless times but is never defined. Please include a sentence or 2 listing the actual property the author is talking about. I give one example on p.7 l.13-15. There I am told there is a positive correlation between freezing properties of microlayer and bulk water. How many properties correlate and what is actually being correlated? Please search for the word and replace it with something that is specific and measurable.

How can the droplet freezing technique analyse videos (p.4 l.11)? That must be automated or done by a person?

SYBR Green stains nucleic acids (p.5 l.25) which means it stains bacteria, phytoplankton, cyanobacteria, archaea and everything biogenic for that matter? The concentration derived from SYBR Green should be subtracted by the phytoplankton counts to get bacteria counts? In addition, there should be other things beside living organisms that stain, for example other biogenic particles such as cell fragments or gel-like particles. Are the authors counting this as well? Is there another name for these counts that should be used?

There is no reason for a 2 sentence long subsection (section 2.5). Please incorporate this elsewhere.

p.6 l.22: It is not nice to the reader to be directed to the supplement for the first result. Please let me read about the main, exciting results first and then take me to those which are supplementary.

p.6 l.27: "In addition,...also..." is repetitive.

p.8 l.16-17: Why is it important to say that similar water masses were samples? The authors sampled from similar locations so why say more? Please tell me what exactly is similar about the water masses besides salinity.

p.10 l.12: The authors did not measure inter-annual variability. They did measure for a month in 2 different years.

**References**

V. E. Irish, P. Elizondo, J. Chen, C. Chou, J. Charette, M. Lizotte, L. A. Ladino, T. W. Wilson, M. Gosselin, B. J. Murray, E. Polishchuk, J. P. D. Abbatt, L. A. Miller and A. K. Bertram, *Atmos. Chem. Phys.*, 2017, **17**, 10583–10595.
T. Koop and B. Zobrist, *Phys. Chem. Chem. Phys.*, 2009, **11**, 10839–10850.

---

## Referee Comment (RC2) · Anonymous Referee #2 · 10 Oct 2018

Irish et al. investigated the ice nucleating particles (INP) in the sea surface microlayer and bulk seawater in the Canadian Arctic during summer of 2014 and 2016. This study measured INP concentrations using the droplet freezing technique. It is also investigated the effects of heat and filtration treatments on the INP concentrations. The manuscript concluded that spatial patterns of INPs are similar between the summers of these two years, but average INP concentrations are higher in 2016 and in some cases, there is INPs enhancement in the microlayer. The manuscript provides a set

of comparison (at the "same" sampling sites) for INP measurements at important geographic location (Arctic) where data are overall limited. The topic of this manuscript is within the scope of this journal. There are some issues and comments should be addressed or considered before it is recommended for the publication.

1. For microlayer samples, the sampling devices and procedures are different when considering what is sampled (the sampling thickness of microlayer). As mentioned in P7/Line 20, how this is contributing to the difference in INP measurements in 2014 and 2016?

2. Justification of using T10 (e.g., why not using T50) for statistical analysis is needed.

3. There is a concern when the manuscript states the equation (1) accounts for the possibility of multiple INPs within a single droplet. It is better to elaborate the point that the authors try to convey.

4. It is not clear how many droplets were investigated for each sample, only 15-30 droplets as stated in P4/Line7?

5. Is there in situ Chl-a measurements which would be more accurate and can be used to correlated to T10?

6. It would be benefit to the community if the manuscript can identify some possible issues when investigating the annual or seasonal variability in INPs over the ocean. This has been done in part in the manuscript, such as the last paragraph.

---

## Author Comment (AC1) · 24 Jan 2019

Prof. Daniel Cziczo
Co-Editor of Atmospheric Chemistry and Physics

Dear Dan,

Listed below are our responses to the comments from the reviewers of our manuscript. For clarity and visual distinction, the referee comments or questions are listed here in black and are preceded by bracketed, italicized numbers (e.g. *[1]*). Authors' responses are in red below each referee statement with matching numbers (e.g. *[A1]*). We thank the reviewers for carefully reading our manuscript and for their helpful suggestions!

Sincerely,

Allan Bertram
Professor of Chemistry
University of British Columbia

**Short Comment #1**

[1] I really enjoyed reading this paper. It is not surprising that there were differences in the IN between years. There could even be differences within a single year. It is important to note that the production mechanism determines how much of the SSML will be transferred to SSA...and there are difs in IN activity of sea spray produced by jet vs film drops (Wang, et al., PNAS, 2017, The role of jet and film drops in controlling the mixing state of submicron sea spray aerosols). The IN efficiency of jet drops has been shown in this publication to be higher than film drops–which goes against the common assumption that there will be more bioparticles in the SSML than in bulk seawater. Further, the # of jet vs film drops can change over the course of a bloom (see Fig 1 below taken from supp info in above referenced PNAS paper). It is important to remember that physical, biological, and chemical factors all control the formation and composition of SSA and will thus affect the IN activities.

*[A1] We thank Professor Kim Prather for her short comment. In the revised manuscript, we will note that the amount of microlayer transferred to sea spray aerosol will depend on the production mechanism, and we will include a reference to the work by Wang et al. (2017).*

**Anonymous Referee #1**

The manuscript titled "Revisiting properties and concentrations of ice nucleating particles in the sea surface microlayer and bulk seawater in the Canadian Arctic during summer" by Irish et al. presents an ice nucleation study using droplets

generated from bulk and surface Arctic seawater. The authors use filtration to estimate the size of the ice nucleating agent. After heating the water, freezing temperatures of droplets decreased. Finally, the authors also measure salinity, numbers of bacteria and phytoplankton cells and correlate them to the temperature at which 10% of the droplets froze, T10, in each sample. Chlorophyll satellite data was also correlated with T10. Warmer freezing temperatures correlated with decreasing salinity and decreasing bacteria concentrations. The authors also found warmer freezing temperatures in this study compared with the one conducted in 2014.

[2] Overall, I find the manuscript unsuitable for ACP in the present form as I fail to see new findings in chemistry and physics. Furthermore, the results and data analysis are essentially identical to Irish et al. (2017) with the only difference of the sampling depth due to use of a rotating drum. Granted, the data here is new because it is taken at a later year and with a well defined scientific approach. However, the results and analysis in this manuscript are essentially copies of the author's previous paper without any scientific extension. In my major comments below, I have introduced some ideas to extend their work. As it is now, I do not find any conclusions anywhere in the manuscript and the word is not even written except for the title of the section. Please understand the difference between conclusions and observational results. Overall, I will not recommend publication in ACP.

*[A2] We thank the referee for pushing us to dig deeper into the data and extend our analysis beyond what was presented in Irish et al. 2017. We also thank the referee for the constructive ideas to extend our work. In response to the referee's comment, we will go back and investigate concentrations of stable isotopes of oxygen to determine the contributions of melting sea ice and terrestrial run-off (including that from melting glaciers and permafrost) to INP concentrations. This data will provide new and interesting insights into the source of the INPs and will be a major focus in the revised manuscript. In addition we will modify Section 4 to clearly state conclusions from the current study. Below we respond to the individual comments from the referee.*

**Major Comments**

[3] The authors lack ice nucleation physics. There is no nucleation theory or any application of active sites for comparison with other studies. This is because there is no surface area estimate of insoluble material in their droplets. If water and filters are still available, then total particulate mass or surface area of insoluble particles could be obtained. For example, filters can be washed, dried and weighed and water can be used to get a size distribution from the flow cytometry. Another point is that correction for freezing point depression follows a water activity approach that Koop and Zobrist (2009) used for other biogenic ice nucleators. A plot of INP vs. $\Delta a_{w,het}$ could be made which allows the authors to

discusses the effect (or lack thereof) of ionic activity on ice nucleation. $\Delta a_{w,het}$ could be compared with other biogenic ice nucleators.

*[A3] Thank you for the suggestion. Unfortunately, the water and filters are no longer available. To address the referee's comments we will include a new section (3.5 Atmospheric implications) where we will normalise our freezing results to the mass of sea salt. In other words, from the measured concentrations of INPs and measured salinities we will calculate the number of INPs per unit mass of sea salt. This data will then be used to extrapolate our measurements to the atmospheric conditions.*

[4] The authors lack cloud and atmospheric physics. The authors could use SSA production formulations measured from previous studies to calculate the number of ice forming particles per liter of air. Vertical and horizontal motion (updraft and 10 m high wind speed) provided from meteorological data, or reanalysis can then be used to give some notion of the total ice nucleating particles in air. Does the T10 data or some other percentage of droplets frozen, correspond to a mixed phase cloud base or ice water path from satellite data?

*[A4] To address the referee's comments, we will include a new section (3.5 Atmospheric implications) where we will estimate the concentrations of INPs in the atmosphere from our measurements.*

[5] The authors lack ocean physics. There is countless studies documenting the enrichment or lack of enrichment of material in a microlayer with respect to bulk water. These materials can be surfactants, insoluble particles, or other materials such as proteins and polysaccharides. The interesting result from both the present manuscript and Irish et al. (2017) is that the ice nucleation ability is the same for bulk and microlayer water. This could mean that the ice nucleating particles are not surface active? What compounds in the ocean are uniformly distributed through the microlayer and bulk water? Are there soluble surfactants in bulk water that are transported to the microlayer? Is there a difference in surface tension between microlayer and bulk water?

*[A5] In the 2016 measurements we saw an enhancement of the concentrations of INPs in the microlayer compared to the bulk in almost 50 % of the samples. We will revise the manuscript to try and make this point clearer.*

[6] p.6 l.26 - p.7 l.1-5: Clearly, freezing temperatures warmer than pure water indicate heterogeneous droplet freezing. However, the "procedural blank" resulted in freezing temperatures at -16∘ C. How is it possible that freezing was observed below this temperature? On l.4-5 the answer is given that rinsing times were different (a fact not mentioned in the experimental section), so the freezing temperatures of the "blank were due to (cross-)contamination. How can we then compare any measurements of these to the blanks? In microlayer samples, Fig. S2a shows that no data below -16∘ C can be trusted. If these were the blanks for

the experiments, the freezing curves should follow exactly the procedural blank data which would be seen as a discontinuity (step in the graph) of the freezing temperature around -16∘ C. This is not the case and so I would conclude that this blank has nothing to do with the data at all and suggest there is no blank experiment for these data using the same procedures. How is this data at all trustable? I now understand why the authors use T10 and not median freezing as reported in Koop and Zobrist (2009), because if they did there would be no difference with their freezing points of microlayer water and the blanks. I am very concerned that freezing temperatures were due to cross contamination because of the lack of reproducibility for the blanks, as the freezing temperatures of the microlayer and bulk seawater do not follow the blanks at all.

*[A6] The referee is correct there is some uncertainty in the microlayer samples that froze at temperatures less than the procedural blanks. As stated in the original manuscript, the freezing temperatures of the procedural blanks should be viewed as an upper limit to the background freezing temperatures, since prior to collecting the blanks, the sampler had not been rinsed as thoroughly as before collecting the microlayer samples. To address the referee's comments, in the revised manuscript, after the difference between the procedural blanks and the samples are first reported, we will only include freezing temperatures for the microlayer that were at warmer temperatures than the procedural blanks. Note, the same conclusions are reached in the manuscript if all the microlayer data are included or if only freezing temperatures warmer than the procedural blanks are included.*

[7] p.6 l.26 - p.7 l.1-5: In the same section I find that freezing temperatures of filtered water (through the sampler) are less than ultra pure water (not through the sampler) by about 5-10∘ C. I doubt the seawater was more pure than the ultrapure water, so what is wrong here? Are the authors certain of the freezing point correction with the E-AIM model? Is there an uncertainty of ±5 − 10∘ C? I cannot accept this result and it makes me seriously doubt the accuracy of these experiments. The blank should be the lowest freezing temperature.

*[A7] The microlayer samples and bulk seawater samples were passed through 0.02 µm filters, whereas, the ultrapure water was only passed through 0.22 µm filters. This difference in pore size can explain the difference in freezing temperatures, as pointed out by the referee. For example, in previous experiments we observed that the freezing temperature of ultrapure water decreases when the water is passed through a 0.02 µm filter compared to a 0.2 µm filter. To address the referee's comments, this information will be added to the revised manuscript.*

[8] Figure S3: The ultrapure water data here is about 5∘ C different from the ultrapure water in Fig. S2. This indicates to me that the authors experiment is reproducible to ±5 ∘ C. This is a large uncertainty which is not stated in the paper.

*[A8] We apologise for this error, and thank the referee for bringing it to our attention. In the original manuscript in Fig. S2 the laboratory blanks correspond to ultrapure water from a MilliQ system in 2016. By mistake, in Fig. S3, we plotted laboratory blanks that correspond to ultrapure water from a MilliQ water system in 2014. The freezing temperatures of the laboratory blanks in 2014 were lower than the freezing temperatures of the laboratory blanks in 2016. This difference is most likely because the UV lamp and filter on our MilliQ water system had been recently changed in 2014, but not in 2016. In the revised manuscript, only the laboratory blanks corresponding to ultrapure water from a MilliQ system in 2016 will be included.*

[9] p.3 l.12: It is not possible to name your instrument as an autosampler when for the majority of the stations the authors had to manually rotate the drum.

*[A9] We will change the name from "automated sampler" to a "sampling catamaran", and "automated sampling" to "sampling" throughout the manuscript.*

[10] Equation 1: How does this equation account for the possibility of multiple INP's? Does the author observe more than one nucleation event in a droplet before it crystalizes? How can they tell if the droplet has 1 or 100 INPs inside? This method of analysis is 45 years old, do the authors have an updated analysis for quantifying freezing?

*[A10] In our experiments, freezing of a single droplet is due to one nucleation event, since after a nucleation event, the freezing time of a droplet is very short (< 1 second). We think the statement in the original manuscript "This equation accounts for the possibility of multiple INPs containing in a single droplet" has led to some confusion. This statement will be removed from the revised manuscript, and additional discussion on Eq. 1 will be added to improve clarity.*

[11] p.8 l.23-31: The logic is flawed here. Melting sea ice decreases salinity and releases bacteria to the ocean (p.24-26). Decreasing salinity yields warmer T10 (Fig. 6 lower left). Decreasing salinity yields increasing bacteria (p.8 l.24-27). Finally, increasing bacteria yield lower T10 (Fig. 6 upper left). So why do I read in p.8 l.30-31 that bacteria are fewer in melting sea ice and that bacteria increasing ice nucleation ability? This argument is highly contradictory.

*[A11] We think the referee misunderstood our argument/logic. To address the referee's comment, we will revise this section and improve clarity.*

[12] p.8 l.23-31: The most major problem I see in the manuscript is that it is a copy of the authors previous manuscript. The majority of section 4 is a repeat of Irish et al. (2017). The last paragraph of section 4 states that the only new finding is that concentrations are higher in 2016 than in 2014, but dismisses this finding due to a different sampler. This study ended in August 2016, but the Irish et al. (2017) paper (using only 2014 data) was submitted April 2017. Why wasn't the

data presented in this manuscript used in Irish et al. (2017)? In any case, the authors should extend their work with new data and new discussion that includes physical and chemical understanding before I recommend publication in ACP.

*[A12] Again, we thank the referee for pushing us to extend our analysis beyond what was presented in Irish et al. (2017). As mentioned above, to address the referee's comment, we will investigate stable isotopes of oxygen to determine the contribution of melting sea ice and terrestrial run-off (including that from melting glaciers and permafrost) to INP concentrations. This data will provide new and interesting insight into the source of the INPs measured. This new data will be a major focus in the revised manuscript.*

**Minor Comments**

[13] What are the "properties" of ice nucleating particles? How is that different from "freezing properties"? How is that different from "ice nucleation properties"? Properties of the microlayer? This word is used countless times but is never defined. Please include a sentence or 2 listing the actual property the author is talking about. I give one example on p.7 l.13-15. There I am told there is a positive correlation between freezing properties of microlayer and bulk water. How many properties correlate and what is actually being correlated? Please search for the word and replace it with something that is specific and measurable.

*[A13] In the revised manuscript we will replace freezing properties and ice nucleation properties with a more specific and measurable term.*

[14] How can the droplet freezing technique analyse videos (p.4 l.11)? That must be automated or done by a person?

*[A14] The authors will change the wording in this sentence to the following:*

*"All videos were analysed manually to determine the freezing temperature of each droplet"*

[15] SYBR Green stains nucleic acids (p.5 l.25) which means it stains bacteria, phytoplankton, cyanobacteria, archaea and everything biogenic for that matter? The concentration derived from SYBR Green should be subtracted by the phytoplankton counts to get bacteria counts? In addition, there should be other things besides living organisms that stain, for example other biogenic particles such as cell fragments or gel-like particles. Are the authors counting this as well? Is there another name for these counts that should be used?

*[A15] Yes, SYBR Green stains all DNA and RNA, but bacteria are easily discriminated from other organisms (or detritus or transparent exopolymeric particles) by their size (side scatter) and fluorescence intensity. In addition,*

*autotrophs stained with SYBR Green are discriminated from heterotrophic bacteria by their chlorophyll a fluorescence. To address the referee's comments this information will be added to the revised manuscript.*

[16] There is no reason for a 2 sentence long subsection (section 2.5). Please incorporate this elsewhere.

*[A16] We will incorporate this sentence into section 3.1.*

[17] p.6 l.22: It is not nice to the reader to be directed to the supplement for the first result. Please let me read about the main, exciting results first and then take me to those which are supplementary.

*[A17] To address the referee's comments, Fig. S2 will be moved from the supplement to the main text.*

[18] p.6 l.27: "In addition,...also..." is repetitive.

*[A18] "In addition" will be removed in the revised manuscript.*

[19] p.8 l.16-17: Why is it important to say that similar water masses were samples? The authors sampled from similar locations so why say more? Please tell me what exactly is similar about the water masses besides salinity.

*[A19] To address the referee's comment, we will remove "which suggests that we sampled similar water masses in both years at those locations".*

[20] p.10 l.12: The authors did not measure inter-annual variability. They did measure for a month in 2 different years.

*[A20] We will remove "inter-annual" to address the referee's comment.*

**References**

V. E. Irish, P. Elizondo, J. Chen, C. Chou, J. Charette, M. Lizotte, L. A. Ladino, T. W. Wilson, M. Gosselin, B. J. Murray, E. Polishchuk, J. P. D. Abbatt, L. A. Miller and A. K. Bertram, Atmos. Chem. Phys., 2017, 17, 10583–10595.
T. Koop and B. Zobrist, Phys. Chem. Chem. Phys., 2009, 11, 10839–10850.

**Anonymous Referee #2**

Irish et al. investigated the ice nucleating particles (INP) in the sea surface microlayer and bulk seawater in the Canadian Arctic during summer of 2014 and 2016. This study measured INP concentrations using the droplet freezing technique. It is also investigated the effects of heat and filtration treatments on the INP concentrations. The manuscript concluded that spatial patterns of INPs

are similar between the summers of these two years, but average INP concentrations are higher in 2016 and in some cases, there is INPs enhancement in the microlayer. The manuscript provides a set of comparison (at the "same" sampling sites) for INP measurements at important geographic location (Arctic) where data are overall limited. The topic of this manuscript is within the scope of this journal. There are some issues and comments should be addressed or considered before it is recommended for the publication.

[21] For microlayer samples, the sampling devices and procedures are different when considering what is sampled (the sampling thickness of microlayer). As mentioned in P7/Line 20, how this is contributing to the difference in INP measurements in 2014 and 2016?

*[21] As stated in line 20, the microlayer samples would be "diluted" by bulk sample. We will try and make this clearer in the revised manuscript.*

[22] Justification of using T10 (e.g., why not using T50) for statistical analysis is needed.

*[22] T10-values were chosen since they are a convenient way to summarise the freezing data and also to be consistent with our previous study. Similar conclusions would be reached in our manuscript if other values (e.g. T50-values were used). In the revised manuscript we will also include T50 values to address the referee's comments.*

[23] There is a concern when the manuscript states the equation (1) accounts for the possibility of multiple INPs within a single droplet. It is better to elaborate the point that the authors try to convey.

*[23] See A10 above.*

[24] It is not clear how many droplets were investigated for each sample, only 15-30 droplets as stated in P4/Line7?

*[24] We will clarify the number of droplets that were investigated for each sample by adding the following to section 2.2.1:*

"In the freezing experiments three hydrophobic glass slides (Hampton Research, Aliso Viejo, CA, USA) were placed directly on a cold stage (Whale et al., 2015) and between 15 to 30 droplets of the sample, with volumes of 1 µL each, were deposited onto each of the glass slides using a pipette. As a result, a total of 45 to 90 droplets were analysed for each sample."

[25] Is there in situ Chl-a measurements which would be more accurate and can be used to correlated to T10?

*[25] To address the referee's comment we will investigate the correlation between T10 and in situ Chl-a measurements, in addition to satellite Chl-a data.*

[26] It would be benefit to the community if the manuscript can identify some possible issues when investigating the annual or seasonal variability in INPs over the ocean. This has been done in part in the manuscript, such as the last paragraph.

*[26] To address the referee's comment, in the Conclusions, we will expand on possible issues in the reported annual or seasonal variability in INPs.*

---

## Referee Report (RR1)

**Review for "Revisiting properties and concentrations of ice nucleating particles in the sea surface microlayer and bulk seawater in the Canadian Arctic during summer" by Victoria E. Irish et al.**

Anonymous Referee

The revised submission of "Revisiting properties and concentrations of ice nucleating particles in the sea surface microlayer and bulk seawater in the Canadian Arctic during summer" by Irish et al. has improved compared to the initial submission, but this is very little to bring it up to the standards of the journal Atmospheric Chemistry and Physics. My previous comments in summary, were that there was insufficient scientific advancement in physics or chemistry to warrant publication. In this submission there are two new extensions beyond what was published in Irish *et al.*[1] which are i) oxygen isotopic fractionation and ii) a quantitative analysis of ice nucleating particles in air from a marine source. The isotope measurements adds to the study that terrestrial runoff and precipitation are correlated with the freezing temperature at which 10% of droplets froze, $T_{10}$. This correlation was better than for melting sea ice or seawater. To calculate the concentration of ice nucleating particles (INPs) in air, mass concentrations of sodium in ambient aerosol were used to scale their results. In total, the new findings compared to Irish *et al.*[1] were that INP concentrations were higher in 2016 than 2014 which are likely due to volume sampling differences, a correlation between calculated meteoric and sea ice melt water fractions and $T_{10}$, and back of the envelope calculations for INP concentrations in air. These extensions unfortunately do not apply any theory or give fundamental understanding in physics and chemistry and so I cannot recommend publication in the journal Atmospheric Chemistry and Physics which stresses exactly this. I warn the authors that if the editor allows for resubmission, much more work must be done in this regard to significantly shorten discussions that are already made in Irish et al. while emphasizing any new discussion. Calculating correlations and up scaling data to the atmosphere is good but not sufficient for greater physical and chemical understanding.

Certainly, the measurements done on board a research vessel are very difficult, and there are now small extensions beyond the previous submission. These should be published, but I recommend elsewhere in literature. I would concede if the authors were restricted in time and submitted some months after

the research cruise was over, then the benefit of the doubt would be given to publish exciting results as soon as possible. Was there some limitation in time or some issue with the data or paper that I should be unaware of when re-evaluating this manuscript?

Major Comments

There remains an absence of testing any theory. This includes any chemistry, physics or thermodynamics. Free energy calculation for ice nucleation or critical ice embryo size is not calculated. Nucleation theories are not applied or tested. There is no evaluation on the transfer of particles from the bulk to the microlayer or into the air that uses physics or chemical transformation. Measurement of biological tracers are done, but only correlation is made without any other hypothesis testing.

The authors did not need to make more clear that they observed enhanced INP numbers in microlayer layer more in 2016 than in 2014 on l. 27-29. They needed to explain and give a physical-chemical reason as to why. Instead they only claim that ocean variability was the cause, or more likely than not it was an artifact of sampling a factor of 3 less in layer thickness 2016. This means that the microlayer concentrations in 2014 were simply diluted. It is true that the authors data make a comparison quantifying how the properties and concentrations of INPs have remained the same or have varied between these years, however, it does not answer the question of why. In general, the authors have not extended their manuscript enough and should choose a different journal that stresses measurements and data more.

The authors state that much of their results and data are consistent with Irish *et al.*[1]. I had previously made the comment that the manuscript was too similar to their previous work, being about 30% identical to Irish *et al.*[1] and other material they published based on the iThenticate.com Similarity Report. Although the addition of oxygen stable isotopes and calculation of airborne INPs will make this less similar, not enough was done to reword the rest of the manuscript. Therefore, my previous major comment that this manuscript it too similar to their previous is still warranted.

Minor Comments

- p.1, l.17 - The word choice is too negative. The way it was in the first version using the word "limited" better states that good work has been done and there is a need for more.

- p.4, l.14-15 - The freezing temperature is not determined visually. The freezing is determined

visualy and the temperature is measured by an instrument at the same time it freezed. Please reword this sentence.

- Please indicate in one sentence or so in section 2.2.1 how temperature was calibrated.

- There is a section 2.1.1 but no section 2.1.2. There is no need to separate here. Please have only section 2.1.

- Description of blanks for the lab and field for different filtering are in different places, p.12 l.30 - p.13 l.2, p.13 l.14 - 16, p.17 l.3-7. Field blanks are discussed many times but found it hard when reading through the paper, where to locate their description. I recommend the authors dedicate a new short section to describe all the blanks one after another. This will help the reader refer back to the definition of the blanks.

- Another point about the field blanks. I understand that when seawater is filtered, freezing temperatures are much lower than field blanks. The procedure to make a field blank is first, to rinse all glassware and tubing for some time then second, sample and freeze drops of pure water that rinsed and flushed all glassware and tubing after the first rinse. Therefore, is it safe to say the purpose for field blanks is to evaluate the ability to reuse the same glass plate sampler and tubing to not cross contaminate between different stations? I think this is the case. It should be directly stated in the manuscript.

- The short sentence on p.6 l.18 should be removed as it is a repeat of the previous.

- The phrase *in situ* was not used in the previous manuscript, but it is used in the revised version. However, an *in situ* chlorophyll measurement was not performed because the authors did not measure in water that remained in the ocean. Water was removed from the ocean. Samples of water were used for chlorophyll concentrations measurements. Please correct this.

- p.9 l.29 - The correlation coefficient of -0.83 and p value of 0.001 is exactly the same for both $T_{10}$ and $T_{50}$ in Tables 2 and S2. Is the a typo or coincidence?

- p.10 l.4-8 - Deviation in freezing temperatures from those of constant $\Delta a_\mathrm{w}$ was observed only for ammonium containing solutes[2]. Ammonia concentration in seawater should be on the order of

micromolar and therefore should not affect freezing temperature in this way. This authors may wish to include this.

- p.10 l.12-14 Terrestrial runoff can also contain nutrients to grow marine microorganisms. After these nutrients are used up, cells can lyse, sink or their exudate can remain in surface waters. Then the source of INP may still be marine organisms. These sentences imply that terrestrial organisms in fresh water/lower salinity water are the major INP source, but this is only one possibility. The authors should include both.

- What does "the upper end of the average values" mean on p.11 l.13? I have never heard of this measure before. Should the authors simply use the average of these 6 values?

- In Fig. 10, there are many conclusions missing that I hope the author would reconsider. First is that similar INP values per volume of air to previous literature is only seen for 2 or 3 stations, at temperatures for -10 to -5 C and more for microlayer samples than seawater samples. Could the authors state that a seawater source of ambient INP should be more important at warmer temperatures than for colder temperatures? At colder temperatures, there may be insignificant contribution of primary emission of INP from seawater. Would their other measurements such as filtering and heat treatment allow for the suggestion that these warm temperature INPs in ambient air may be from primary emission and also biogenic? Can the authors claim any evidence for a known aerosolized biogenic particle in the size range of $0.02 - 0.2$ $\mu$m? Is algal and bacterial exudate this size?

**References**

[1] V. E. Irish, P. Elizondo, J. Chen, C. Chou, J. Charette, M. Lizotte, L. A. Ladino, T. W. Wilson, M. Gosselin, B. J. Murray, E. Polishchuk, J. P. D. Abbatt, L. A. Miller and A. K. Bertram, *Atmos. Chem. Phys.*, 2017, **17**, 10583–10595.

[2] A. Kumar, C. Marcolli, B. Luo and T. Peter, *Atmospheric Chemistry and Physics*, 2018, **18**, 7057–7079.

---

## Author Response (AR2)

Prof. Daniel Cziczo
Co-Editor of Atmospheric Chemistry and Physics

Dear Dan,

Listed below are our responses to the second round of comments from Reviewer #1 of our manuscript. For clarity and visual distinction, the reviewer's comments or questions are listed here in black and are preceded by bracketed, italicized numbers (e.g. *[1]*). Authors' responses are in red below each referee statement with matching numbers (e.g. *[A1]*). We thank the reviewer for carefully reading our manuscript and providing feedback to improve our manuscript.

Sincerely,

Allan Bertram
Professor of Chemistry
University of British Columbia

**Review for "Revisiting properties and concentrations of ice nucleating particles in the sea surface microlayer and bulk seawater in the Canadian Arctic during summer" by Victoria E. Irish et al.**

**Anonymous Referee**

The revised submission of "Revisiting properties and concentrations of ice nucleating particles in the sea surface microlayer and bulk seawater in the Canadian Arctic during summer" by Irish et al. has improved compared to the initial submission, but this is very little to bring it up to the standards of the journal Atmospheric Chemistry and Physics. My previous comments in summary, were that there was insufficient scientific advancement in physics or chemistry to warrant publication. In this submission there are two new extensions beyond what was published in Irish et al.[1] which are i) oxygen isotopic fractionation and ii) a quantitative analysis of ice nucleating particles in air from a marine source. The isotope measurements adds to the study that terrestrial runoff and precipitation are correlated with the freezing temperature at which 10% of droplets froze, T10. This correlation was better than for melting sea ice or seawater. To calculate the concentration of ice nucleating particles (INPs) in air, mass concentrations of sodium in ambient aerosol were used to scale their results. In total, the new findings compared to Irish et al.[1] were that INP concentrations were higher in 2016 than 2014 which are likely due to volume sampling differences, a correlation between calculated meteoric

and sea ice melt water fractions and T10, and back of the envelope calculations for INP concentrations in air. These extensions unfortunately do not apply any theory or give fundamental understanding in physics and chemistry and so I cannot recommend publication in the journal Atmospheric Chemistry and Physics which stresses exactly this. I warn the authors that if the editor allows for resubmission, much more work must be done in this regard to significantly shorten discussions that are already made in Irish et al.[1] while emphasizing any new discussion. Calculating correlations and up scaling data to the atmosphere is good but not sufficient for greater physical and chemical understanding.

Certainly, the measurements done on board a research vessel are very difficult, and there are now small extensions beyond the previous submission. These should be published, but I recommend elsewhere in literature. I would concede if the authors were restricted in time and submitted some months after the research cruise was over, then the benefit of the doubt would be given to publish exciting results as soon as possible. Was there some limitation in time or some issue with the data or paper that I should be unaware of when re-evaluating this manuscript?

**Major Comments**

**[1]** There remains an absence of testing any theory. This includes any chemistry, physics or thermodynamics. Free energy calculation for ice nucleation or critical ice embryo size is not calculated. Nucleation theories are not applied or tested. There is no evaluation on the transfer of particles from the bulk to the microlayer or into the air that uses physics or chemical transformation. Measurement of biological tracers are done, but only correlation is made without any other hypothesis testing.

The authors did not need to make more clear that they observed enhanced INP numbers in microlayer layer more in 2016 than in 2014 on l. 27-29. They needed to explain and give a physical-chemical reason as to why. Instead they only claim that ocean variability was the cause, or more likely than not it was an artifact of sampling a factor of 3 less in layer thickness 2016. This means that the microlayer concentrations in 2014 were simply diluted. It is true that the authors data make a comparison quantifying how the properties and concentrations of INPs have remained the same or have varied between these years, however, it does not answer the question of why. In general, the authors have not extended their manuscript enough and should choose a different journal that stresses measurements and data more.

*[A1] Based on the Atmospheric Chemistry and Physics (ACP) website, ACP is dedicated to the publication and public discussion of high-quality studies investigating the Earth's atmosphere and the underlying chemical and physical processes. It covers the altitude range from the ocean surface up to the tropopause. The current study focuses on the properties and*

*concentrations of ice nucleating particles (INPs) in the sea surface microlayer and bulk seawater in the Arctic. We think this topic is relevant for ACP, since INPs from the Arctic Ocean are potentially an important source of INPs to the atmosphere and because the current manuscript includes estimates of concentrations of INPs in the Arctic marine boundary layer.*

**[2]** The authors state that much of their results and data are consistent with Irish et al.[1]. I had previously made the comment that the manuscript was too similar to their previous work, being about 30% identical to Irish et al.[1] and other material they published based on the iThenticate.com Similarity Report. Although the addition of oxygen stable isotopes and calculation of airborne INPs will make this less similar, not enough was done to reword the rest of the manuscript. Therefore, my previous major comment that this manuscript it too similar to their previous is still warranted.

*[A2] We have gone back and modified text that was similar with Irish et al. 2017. After these modifications, 17% of the wording in the current document is identical to Irish et al. 2017, based on the Similarity Report from Copyleaks.com. However, almost all this overlap is due to similar references in both documents, which cannot be avoided because both manuscripts investigate INPs from the sea surface microlayer and bulk seawater. In addition, in the revised manuscript we have moved Figures 4 and 7 to the Supplement so that the figures in the revised manuscript only emphasize new findings.*

**Minor Comments**

[3] • p.1, l.17 - The word choice is too negative. The way it was in the first version using the word "limited" better states that good work has been done and there is a need for more.

*[A3] We have changed the wording back to "limited".*

[4] • p.4, l.14-15 - The freezing temperature is not determined visually. The freezing is determined visualy and the temperature is measured by an instrument at the same time it freezed. Please reword this sentence.

*[A4] We have re-worded this sentence to the following:*

"The freezing temperature of each droplet was determined from the recorded videos and the temperature history of the cold stage." (Whale et al., 2015)."

[5] Please indicate in one sentence or so in section 2.2.1 how temperature was calibrated.

*[A5] We have added the following sentence to indicate how temperature was calibrated:*

"The temperature of the cold stage was calibrated by measuring the melting temperatures of dodecane (-9.57 °C) and octanol (-14.8 °C) (Whale et al., 2015)."

[6] • There is a section 2.1.1 but no section 2.1.2. There is no need to separate here. Please have only section 2.1.

*[A6] We have made the suggested changes.*

[7] • Description of blanks for the lab and field for different filtering are in different places, p.12 l.30 -p.13 l.2, p.13 l.14 - 16, p.17 l.3-7. Field blanks are discussed many times but found it hard when reading through the paper, where to locate their description. I recommend the authors dedicate a new short section to describe all the blanks one after another. This will help the reader refer back to the definition of the blanks.

*[A7] As suggested we have added a new short section to describe all the blanks one after another (see Section 2.2.2).*

[8] • Another point about the field blanks. I understand that when seawater is filtered, freezing temperatures are much lower than field blanks. The procedure to make a field blank is first, to rinse all glassware and tubing for some time then second, sample and freeze drops of pure water that rinsed and flushed all glassware and tubing after the first rinse. Therefore, is it safe to say the purpose for field blanks is to evaluate the ability to reuse the same glass plate sampler and tubing to not cross contaminate between different stations? I think this is the case. It should be directly stated in the manuscript.

*[A8] Yes, field blanks were used to assess cross contamination between different stations. This information has been added to the revised manuscript.*

[9] • The short sentence on p.6 l.18 should be removed as it is a repeat of the previous.

*[A9] This short sentence has been removed.*

[10] • The phrase in situ was not used in the previous manuscript, but it is used in the revised version. However, an in situ chlorophyll measurement was not performed because the authors did not measure in water that remained in the ocean. Water was removed from the ocean. Samples of water were used for chlorophyll concentrations measurements. Please correct this.

*[A10] We have removed the term "in situ" from the manuscript.*

[11] • p.9 l.29 - The correlation coefficient of -0.83 and p value of 0.001 is exactly the same for both T10 and T50 in Tables 2 and S2. Is the a typo or coincidence?

*[A11] This is coincidence.*

[12] • p.10 l.4-8 - Deviation in freezing temperatures from those of constant _aw was observed only for ammonium containing solutes[2]. Ammonia concentration in seawater should be on the order of micromolar and therefore should not affect freezing temperature in this way. This authors may wish to include this.

*[A12] We have added this information to the revised manuscript.*

[13] • p.10 l.12-14 Terrestrial runoff can also contain nutrients to grow marine microorganisms. After these nutrients are used up, cells can lyse, sink or their exudate can remain in surface waters. Then the source of INP may still be marine organisms. These sentences imply that terrestrial organisms in fresh water/lower salinity water are the major INP source, but this is only one possibility. The authors should include both.

*[A13] Good suggestion. In the revised manuscript we have also discussed the possibility that terrestrial runoff could enhance the production of INPs in the ocean by providing nutrients for the growth of marine microorganisms.*

[14] • What does "the upper end of the average values" mean on p.11 l.13? I have never heard of this measure before. Should the authors simply use the average of these 6 values?

*[A14] To address the referee's comment, we have changed "which is at the upper end of the average values mentioned above" to "which is within the range of the concentrations mentioned above".*

[15] • In Fig. 10, there are many conclusions missing that I hope the author would reconsider. First is that similar INP values per volume of air to previous literature is only seen for 2 or 3 stations, at temperatures for -10 to -5 C and more for microlayer samples than seawater samples. Could the authors state that a seawater source of ambient INP should be more important at warmer temperatures than for colder temperatures? At colder temperatures, there may be insignificant contribution of primary emission of INP from seawater. Would their other measurements such as filtering and heat treatment allow for the suggestion that these warm temperature INPs in ambient air may be from primary emission and also biogenic? Can the authors claim any evidence for a known aerosolized biogenic particle in the size range of 0.02−0.2 µm? Is algal and bacterial exudate this size?

*[A15]  To address the referee's comments, we have expanded our discussion of Fig. 10. Specifically we have included the following:*

"Based on our freezing data, $[INP(T)]_{vol,air}$ ranges from ~$10^{-4}$ L$^{-1}$ to < $10^{-6}$ L$^{-1}$ for freezing temperatures ranging from -5 °C to -10 °C. Over this temperature range, the highest estimated values for $[INP(T)]_{vol,air}$ were associated with two microlayer samples, and only these two microlayer samples resulted in $[INP(T)]_{vol,air}$ values as high as observed in direct atmospheric measurements of $[INP(T)]_{vol,air}$ in the marine boundary layer (Fig. 10) (DeMott et al., 2016; Irish et al., 2019). For freezing temperatures ranging from -15 °C to -25 °C, our estimated $[INP(T)]_{vol,air}$ values range from >$10^{-4}$ L$^{-1}$ to < $10^{-6}$ L$^{-1}$. Over this temperature range, many of our samples result in $[INP(T)]_{vol,air}$ values much less than observed in direct atmospheric measurements (Fig. 10). However, since our estimated $[INP(T)]_{vol,air}$ values are limited to $\lesssim 2 \times 10^{-6}$ L$^{-1}$, we cannot determine if our most active samples result in $[INP(T)]_{vol,air}$ values similar to direct atmospheric measurements for freezing temperatures of -15 °C to -25 °C."

**Comment [13]:** Addresses [13]

**3.4.1 Chlorophyll *a* correlations**

Figure 7 shows correlations between the chlorophyll data retrieved from GlobColour and the $T_{10}$-values for the microlayer and bulk seawater. The correlations between $T_{10}$-values in the microlayer or bulk seawater and chlorophyll *a* are not statistically significant. Figure S9 shows the relationship between the measured chlorophyll *a* concentrations and the $T_{10}$-values for the microlayer and bulk seawater. Again, the correlations are not statistically significant. Our results from satellite and our measured chlorophyll *a* data are consistent with recent work by Wang et al. (2015), who showed that INP concentrations in sea spray aerosol emitted during a mesocosm tank experiment were not simply coupled to chlorophyll *a* concentrations.

**3.5 Predictions of INP concentrations in the Arctic marine boundary layer**

In the following, we provide an initial estimate of the concentration of INPs in the Arctic marine boundary layer based on our freezing results. First, we calculated the concentration of INPs in the liquid per unit mass of sea salt, $[INP(T)]_{mass,salt}$, using the following equation:

$$[INP(T)]_{mass,salt} = -\ln\left(\frac{N_u(T)}{N_o}\right)N_o \cdot \frac{1}{V\rho S} \qquad (6)$$

where $\rho$ is the density of water and $S$ is the salinity of the seawater. Plots of $[INP(T)]_{mass,salt}$ as a function of temperature calculated from our freezing results are shown in Fig. S10. From $[INP(T)]_{mass,salt}$, the concentration of INPs per unit volume of air in the Arctic marine boundary layer, $[INP(T)]_{vol,air}$ was estimated with following equation:

$$[INP(T)]_{vol,air} = [INP(T)]_{mass,salt} \cdot c \qquad (7)$$

where $c$ is the concentration of sea salt in the Arctic marine boundary layer. Average concentrations of sea salt at Barrow, Alaska (71.3° N, 156.6° W), Alert, Nunavut, Canada (82.5° N, 62.5° W), and Zeppelin, Svalbard, Norway (78.9° N, 11.9° E) are 1.5, 0.1, and 0.6 $\mu g\ m^{-3}$ in July, and 1.4, 0.1, and 0.5 $\mu g\ m^{-3}$ in August, respectively (Huang et al., 2018a). For these exploratory calculations we used a value of 1 $\mu g\ m^{-3}$, which is within the range of the concentrations mentioned above.

Shown in Fig. 8 are estimated values for $[INP(T)]_{vol,air}$ based on our freezing data and a concentration of sea salt in the Arctic marine boundary layer of 1 $\mu g\ m^{-3}$. Based on our freezing data, $[INP(T)]_{vol,air}$ ranges from ~$10^{-4}$ $L^{-1}$ to < $10^{-6}$ $L^{-1}$ for freezing temperatures ranging from -5 °C to -10 °C. Over this temperature range, the highest estimated values for $[INP(T)]_{vol,air}$ were associated with two microlayer samples, and only these two microlayer samples resulted in $[INP(T)]_{vol,air}$ values as high as observed in direct atmospheric measurements of $[INP(T)]_{vol,air}$ in the marine boundary layer (Fig. 8) (DeMott et al., 2016; Irish et al., 2019). For freezing temperatures ranging from -15 °C to -25 °C, our estimated $[INP(T)]_{vol,air}$ values range from >$10^{-4}$ $L^{-1}$ to < $10^{-6}$ $L^{-1}$. Over this temperature range, many of our samples result in $[INP(T)]_{vol,air}$ values much less than observed in direct atmospheric measurements (Fig. 8). However, since our estimated $[INP(T)]_{vol,air}$ values are limited to 2 × $10^{-6}$ $L^{-1}$, we cannot determine if our most active samples give $[INP(T)]_{vol,air}$ values

Victoria Irish 2019-4-29 5:39 PM
**Comment [14]:** Addresses [10]

Victoria Irish 2019-4-22 7:33 PM

Victoria Irish 2019-4-29 5:39 PM
**Comment [15]:** Addresses [10]

Victoria Irish 2019-4-22 7:34 PM

Victoria Irish 2019-4-29 5:40 PM
**Comment [16]:** Addresses [14]

[revised manuscript text omitted]